# The dynamics of plasmon-induced hot carrier creation in colloidal gold

Anna Wach [1,2,3], Robert Bericat-Vadell [4], Camila Bacellar [2], Claudio Cirelli [2], Philip J. M. Johnson [2], Rebeca G. Castillo[5], Vitor R. Silveira [4], Peter Broqvist[6], Jolla Kullgren[6], Alexey Maximenko[1], Tomasz Sobol[1], Ewa Partyka-Jankowska[1], Peter Nordlander [7,8], Naomi J. Halas [7,8,9], Jakub Szlachetko[1] ✉ & Jacinto Sá [3,4] ✉

The generation and dynamics of plasmon-induced hot carriers in gold nanoparticles offer crucial insights into nonequilibrium states for energy applications, yet the underlying mechanisms remain experimentally elusive. Here, we leverage ultrafast X-ray absorption spectroscopy (XAS) to directly capture hot carrier dynamics with sub-50 fs temporal resolution, providing clear evidence of plasmon decay mechanisms. We observe the sequential processes of Landau damping (~25 fs) and hot carrier thermalization (~1.5 ps), identifying hot carrier formation as a significant decay pathway. Energy distribution measurements reveal carriers in non-Fermi-Dirac states persisting beyond 500 fs and observe electron populations exceeding single-photon excitation energy, indicating the role of an Auger heating mechanism alongside traditional impact excitation. These findings deepen the understanding of hot carrier behavior under localized surface plasmon resonance, offering valuable implications for applications in photocatalysis, photovoltaics, and phototherapy. This work establishes a methodological framework for studying hot carrier dynamics, opening avenues for optimizing energy transfer processes in nanoscale plasmonic systems.

Surface plasmons, the collective oscillations of conduction electrons in metallic nanostructures, are recognized as an essential elementary excitation in condensed matter, giving rise to multiple practical applications. They can capture far-field radiation and focus it within sub-wavelength regions, defying diffraction limits[1,2], resulting in potent near-fields and significant field amplifications[3]. These characteristics have propelled innovative applications of plasmonics, such as highly sensitive biosensing[4], photothermal therapy for cancer[5], photovoltaics[6,7], and photocatalysis[8].

Surface plasmons exhibit finite lifetimes, decaying either by photon emission (radiatively) or the creation of electron-hole pairs (nonradiatively). Over the past decade, the radiative decay pathway has been researched extensively, yielding the development of efficient nanoantennas that amplify and steer emission from individual emitters[9,10]. Recent research has focused on leveraging nonradiative decay for applications[11]. Hot carriers can initiate chemical reactions in adjacent molecules, even those that demand high-energy under conventional thermal conditions[12,13]. Moreover, plasmon-induced hot

[1]SOLARIS National Synchrotron Radiation Centre, Jagiellonian University, Krakow, Poland. [2]Paul Scherrer Institut, Villigen PSI, Switzerland. [3]Institute of Physical Chemistry, Polish Academy of Sciences, Warsaw, Poland. [4]Department of Chemistry-Ångström, Physical Chemistry division, Uppsala University, Uppsala, Sweden. [5]Max Planck Institute for Chemical Energy Conversion, Mülheim an der Ruhr, Mülheim an der Ruhr, Germany. [6]Maxepartment of Chemistry-Ångström, Structural Chemistry division, Uppsala University, Uppsala, Sweden. [7]Department of Electrical and Computer Engineering, Rice University, Houston, TX, USA. [8]Department of Physics and Astronomy, Rice University, Houston, TX, USA. [9]Department of Chemistry, Rice University, Houston, TX, USA. ✉e-mail: jakub.szlachetko@uj.edu.pl; jacinto.sa@kemi.uu.se

carriers offer a potent means to transform light into electrical currents[14], fostering original solar energy converters[15] and circumventing the bandgap limitations of traditional photodetectors[16].

While the direct excitation of hot carriers on metal surfaces using high-intensity laser pulses has been a longstanding practice in surface femtochemistry, exploiting surface plasmon decay to intensify hot carrier generation is a recent development. This significant advance stems from the remarkably boosted light-harvesting ability of collective plasmon excitations, combined with the substantial enhancement of the plasmon-induced field when metals are nano-confined. Comprehending the underlying physical mechanisms driving plasmon-induced hot carrier generation is essential to fully leveraging these benefits. Although theoretical frameworks elucidating this phenomenon exist[17–22], suitable experimental methodologies are still needed to validate these models.

X-ray absorption spectroscopy (XAS) provides a way to investigate the interactions between X-ray photons and matter, simultaneously providing insight into a material's electronic and chemical characteristics. When X-ray photons are directed toward a material, they can be absorbed by core electrons, resulting in these electrons' transitions to higher energy states. The precise energy at which this absorption occurs depends on the specific material's electronic structure and local environment, making the technique element-specific and highly sensitive.

Transient XAS (or time-resolved XAS (TR-XAS)) probes empty states around the Fermi energy and, in the case of $d^{10}$ metals with the $L_3$-edge transition, provides direct information on the number of carriers involved in electronic transitions and their nonequilibrium energy distributions[23]. At synchrotrons, such dynamical measurements are typically hampered by limited temporal resolution (~50-100 ps) and photon density[24], impeding real-time observations of the hot carrier generation process[8]. However, this limitation has been surpassed by the advent of hard X-ray free electron lasers (XFELs)[25], capable of delivering intense and ultrashort hard X-ray pulses (up to 30 keV at the European XFEL and 12 keV at SwissFEL (used in this study) of less than 50 fs in duration[26]. With this combination of high photon energies and ultrashort pulses, TR-XAS has become an exceptionally valuable experimental probe of dynamical processes. Typical time-resolved measurements are implemented in a pump-probe scheme, where an optical-frequency pump laser triggers electron dynamics, and the X-ray probe captures the evolving nonequilibrium electron distribution. Over the past few years, femtosecond TR-XAS studies have been used to probe photoinduced electronic and structural changes in photoexcited transition metal oxides[27–30] and complexes[31]. In this study, TR-XAS was used to observe the generation and relaxation of plasmon-induced hot carriers in gold nanoparticles (Au NPs) directly because this is an element-specific technique with sufficient temporal resolution. Hot carriers emerge from the interaction between external electric fields and valence electrons, creating electrons and holes with energies above and below the Fermi level ($E_F$). Notably, previous attempts from Bigot et al[32]. and Lehmann et al[33]. with femtosecond optical pump-probe investigations with ionising probe pulses provided earlier evidence for hot electrons and their dynamics but could provide no information about hot holes. On the other hand, Pelli Cresi et al[34]. investigated the electron transfer process in a hybrid plasmonic/semiconductor system (Ag/CeO$_2$) following photoexcitation of the LSPR in the silver NPs by time-resolved soft X-ray absorption spectroscopy. Their findings reveal that the electronic structure of the cerium atoms undergoes an ultrafast change within the first few hundred femtoseconds and persists for at least up to about 1 ps delay time. Their work focused on the plasmon-mediated charge transfer process, however, it did not provide information about hot carrier formation.

## Results and discussion
### Plasmonic optical response and hot carriers temperature
The mechanism for hot carrier formation and thermalization following localized surface plasmon resonance (LSPR) excitation is illustrated in

Fig. 1A[8,22]. In brief, the light's electric field coherently excites the valence electrons in gold, and the subsequent decoherence of this plasmonic excitation leads to non-thermal electron distributions. This occurs via intraband transitions, often aided by phonon scattering or transitions from Landau damping and surface collisions[22], with a time scale of 10–100 fs[8,21]. Initially, photon absorption produces an out-of-equilibrium electron energy distribution, which resembles a double-step function in electron occupancy. Over hundreds of femtoseconds, energy redistribution among the electrons progresses until a high-temperature Fermi-Dirac distribution is reached. Finally, electron-phonon interactions lower the electron temperature over the course of a few picoseconds. This process has been well-supported by theoretical models[35–39] and confirmed experimentally through pump-probe optical spectroscopy[40–45], wherein a visible/near-infrared pump excites conduction band carriers, and a probe pulse tracks the time-dependent changes in transmission or reflectivity as the carriers generate and relax. The non-radiative decay of the plasmon resonance further relaxes through phonon-phonon scattering over hundreds of picoseconds, eventually releasing the generated heat to the surroundings over tens of nanoseconds. These latter stages are beyond the scope of this study.

Au NPs with an average particle diameter of $6.8 \pm 0.9$ nm were used, as confirmed by transmission electron microscopy (TEM, Figs. 1B, S1) and dynamic light scattering (DLS) measured in H$_2$O (solvent) (Fig. S2). The Au NPs have an LSPR centered at 520 nm (2.38 eV) according to UV-Vis spectroscopy (Fig. S3), consistent with previous reports[46]. The UV-Vis spectra of the solution had an optical absorption at the LSPR maximum of ~ 0.2, which was used in conjunction with the DLS to validate the homogeneity of concentration and particle size of the colloidal solutions used in the experiments. Transient absorption spectroscopy (TAS) data were acquired using pump-probe methodology, with an excitation (pump) at 535 nm (-2.32 eV) and a white light (400–800 nm) as a probe. A 5 ×10$^{-3}$M Au NPs aqueous solution was excited with a pump pulse duration of approximately 40 fs and a maximum power density of 1.67 mJ/cm$^2$.

The TAS data are shown in Fig. 1C. Laser excitation of LSPR using an ultrashort pump pulse initiates the generation of excited electrons and holes above and below the Fermi level. Notably, under standard laser fluences, numerous excited carriers are produced, collectively inducing a bleaching effect on the plasmon resonance observed in the transient absorption spectrum (see Fig. 1C)[47]. This initial alteration in electron occupation triggered by the pump pulse is usually referred to as a nonthermal distribution, which subsequently transitions through electron-electron scattering into a thermally equilibrated distribution following Fermi-Dirac statistics[48]. The initial electron-electron scattering contributes to the rising phase of the transient absorption signal, which subsequently diminishes as electron-phonon coupling commences, leading to the equalization of electron and lattice temperatures within a few picoseconds. Electron-phonon coupling plays a crucial role in lowering the electron temperature due to the difference in the heat capacities of electrons and the lattice. Subsequent cooling towards room temperature occurs via phonon-phonon coupling with the surroundings ($\tau_{ph-ph}$), typically within hundreds of picoseconds[46]. The primary decay constant ($\tau_{e-ph}$) measured by ultrafast transient absorption spectroscopy for Au NPs corresponds to electron–phonon coupling, which happens when electrons and holes are still "electronically hot" rather than just "thermally hot"[46]. This term can be used to estimate the average electron temperature, as outlined in the following sections. The kinetic traces extracted at 500 nm were fitted with a double-exponential decay model to obtain the $\tau_{e-ph}$ for the short-lived electron-phonon scattering component and $\tau_{ph-ph}$ for the longer-lived phonon-phonon scattering component, as described in the supplementary information (SI).

The relationship of the TAS data with excitation power is evaluated by fitting kinetic traces extracted in the bleach region, in this

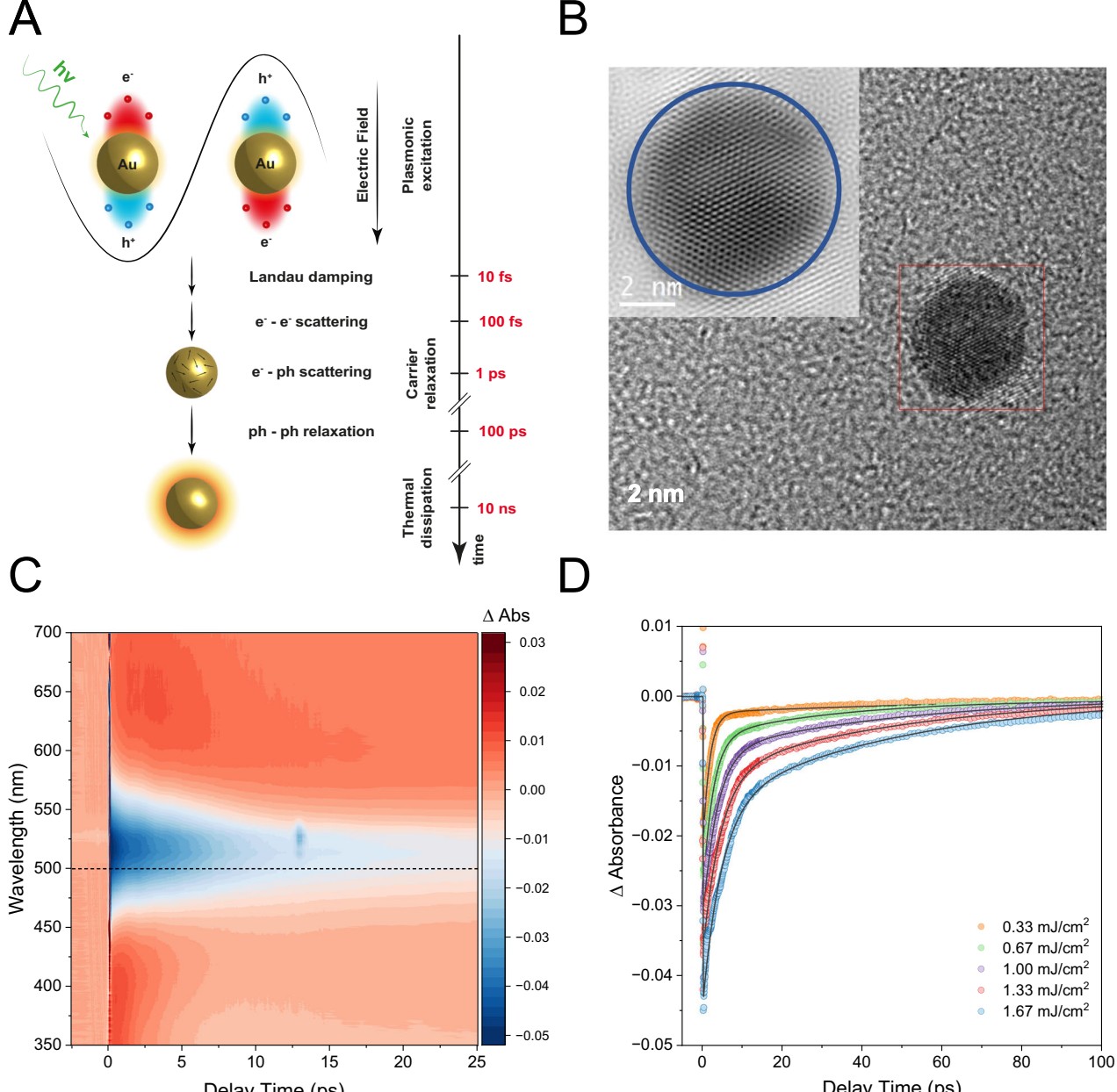

**Fig. 1 | Ultrafast carrier dynamics of gold nanoparticles (NPs) gained by optical spectroscopy. A** Illustration of the theorised plasmonic resonance decay mechanism, including hypothesized time constants for each process. **B** Transmission electron microscopy (TEM) image of a characteristic Au NP of this study. Inset shows a higher-resolution TEM image, revealing the particle's high crystallinity. **C** Transient absorption spectroscopy of Au NPs excited at 535 nm with 1.67 mJ/cm², depicting the characteristic bleach signal and the two positive winglets. **D** Excitation power-dependent bleach recovery dynamics. Kinetic traces extracted at 500 nm (horizontal dashed line in (**C**) with double-exponential fits, shown across varying laser fluences.

case, at 500 nm (Fig. 1D). This relationship can be understood through the two-temperature model[49], which describes the concurrent changes in electron ($T_e$) and lattice ($T_l$) temperatures over time ($t$) by coupled differential equations. This interaction is governed by the electron-phonon coupling constant ($g$)[50–53]:

$$C_e(T_e)\frac{dT_e}{dt} = -g(T_e - T_l) \; and \; C_l(T_l)\frac{dT_l}{dt} = +g(T_e - T_l) \quad (1)$$

Here, $C_e$ and $C_l$ are the electron and lattice heat capacities. Importantly, $C_e$ is temperature-dependent according to $C_e(T_e) = \gamma T_e$, where $\gamma$ is the electron heat capacity constant. For bulk gold, $\gamma = 66 \; Jm^{-3}K^{-2}$ and $g = 2.5 \pm 0.5 \times 10^{16} \; Wm^{-3}K^{-1}$[50]. The variation in electron heat capacity

with temperature leads to the observed dependence on excitation power because the excitation power modulates the initial electron temperature. When the increase in electron temperature remains modest, the linear relationship of electron heat capacity with temperature persists[36,54]. Consequently, the interconnected equations can be reformulated to establish an electron-phonon relaxation time ($\tau_{e-ph}$):

$$\tau_{e-ph} = \frac{\gamma(T_0 + \Delta T_e)}{g} \quad (2)$$

Here, $T_0$ is the ambient temperature (291 K) and $\Delta T_e$ is the pump-induced temperature change of the electrons. According to Eq. 2, the

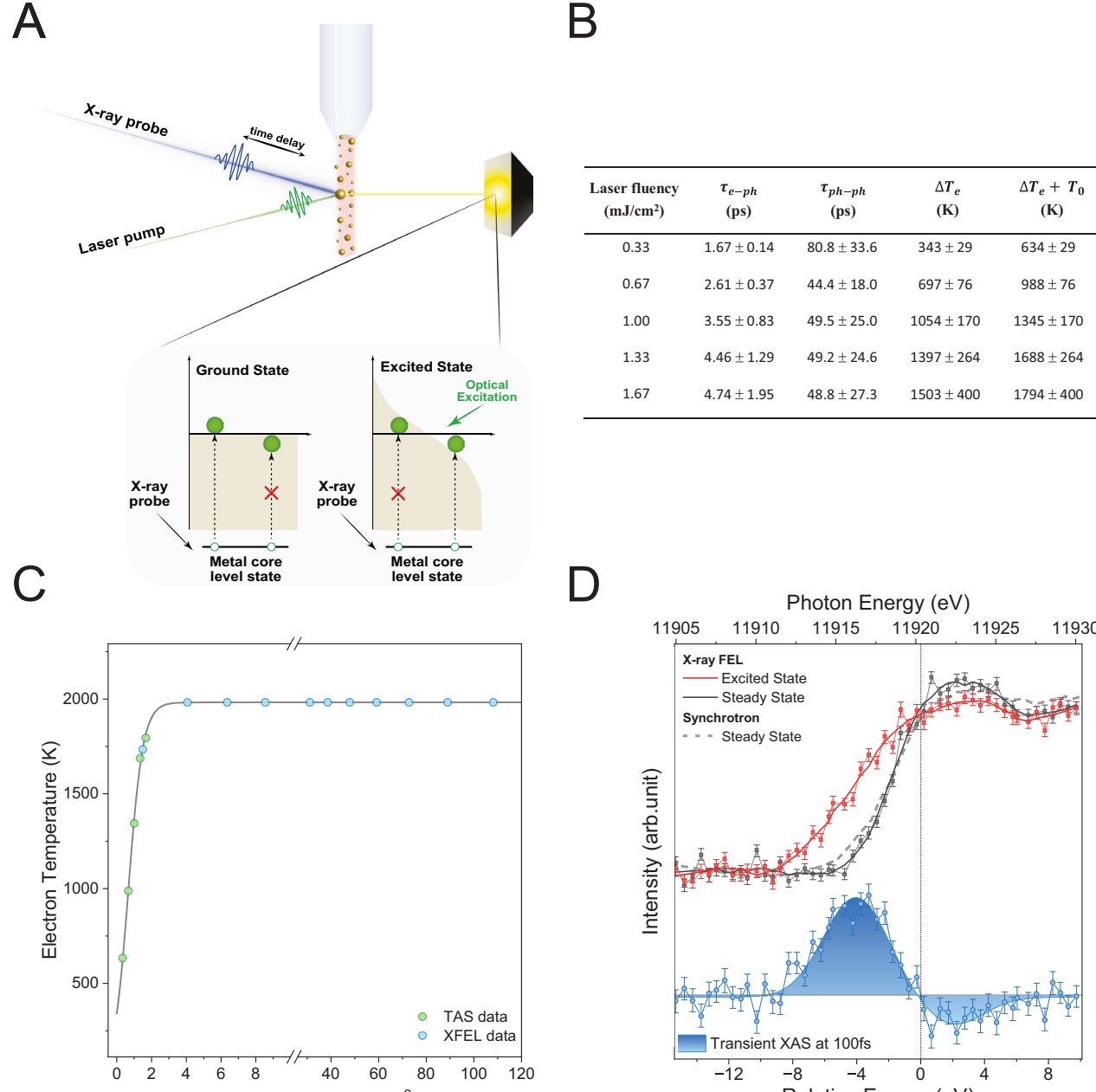

**Fig. 2 | X-ray absorption signatures of gold nanoparticles. A** Au NPs hot carrier generation, multiplication and thermalisation were investigated using pump-probe ultrafast transient XANES with colloidal nanoparticles in water, circulated in a liquid jet. The expected changes due to optical excitation are schematically represented. **B** Tabulation of the best fitting parameters for the LSPR decay channels. **C** Calculated change in the electron temperature versus laser fluence. The TAS data is combined with the TR-XAS signal at an incident energy of 11916 eV and a pump-probe delay of 100 fs. **D** Superimposed $L_3$-edge spectra of steady-state (black trace) and excited-state (red trace) Au NPs with the excited spectrum recorded at $\Delta t = 100$ fs time delay after excitation at 532 nm. The transient XAS spectrum (blue trace) is the difference between excited (pumped) and steady-state (unpumped) spectra. A positive signal in the difference spectrum equates to an increase in empty states (holes) and vice versa. The steady-state XAS spectrum of the Au NPs measured at the synchrotron (grey, dashed line) is shown for comparison.

$\Delta T_e$ was estimated for each laser fluence from the $\tau_{e-ph}$, extracted from the first exponential decay of the plasmonic resonance TAS data. This analysis is summarized in the Table in Fig. 2B. It is noticeable that the increase in laser fluence leads to a rise in the $\tau_{e-ph}$ and, consequently, the $\Delta T_e$, as anticipated[53]. However, it is also observed that at high fluencies, the linearity of the process starts to disappear and the $\Delta T_e$ saturates (Fig. 2C). This is consistent with the two-temperature model predictions, which establish the electron temperature threshold for gold to be lower than $\sim 3000$ K[36,55]. In practice, this means that further increases in fluence primarily affect the number of hot carriers rather than their energy. This temperature analysis

demonstrates that the pump fluence used at the XFEL falls within this stagnation regime, leading to the generation of many hot carriers rather than an increase in carrier energy. This insight is critical for understanding the population of hot carriers under these experimental conditions and provides a basis for analyzing the subsequent electron relaxation dynamics.

The limitation of transient optical measurements is that the nonthermal and thermal carrier populations in plasmonic systems are merely inferred by making assumptions about the functional form of the initial energy distribution or by using indirect monitoring methods, such as localized plasmon frequency shifts[56]. Fig. 2A illustrates the

TR-XAS approach for tracking the changes in the metal density of states (DOS) induced by LSPR excitation, i.e., the direct detection of hot carrier energy distribution. This study specifically focuses on the X-ray absorption near edge structure (XANES) part of the XAS spectrum, which directly monitors the electronic changes in the material, i.e., information on LSPR-induced hot carrier formation. The transient data was collected using the analogous pump-probe methodology that was adopted for the TAS measurements. The technique involves "pumping" the sample with an optical laser pulse, then "probing" with an XFEL fs X-ray pulse. Transient data is acquired by varying the relative time delay between optical and X-ray pulses. To prevent the excitation of damaged Au NPs induced by intense XFEL pulses, a liquid jet was employed to circulate the Au NPs suspended in water. The colloidal solution was refreshed every four hours. The experiments were performed in a climate-controlled laboratory, which, in conjunction with sample circulation (i.e., reduction of local heat deposition), ensured that experiments were performed under isothermal conditions.

## Ultrafast dynamics of plasmonic hot carriers

Au NPs are commonly utilized as ideal test models in XFEL studies on diffraction and imaging[57]. However, to our knowledge, ultrafast XAS studies on Au nanoparticles have yet to be conducted at these facilities, partly because access to the hard X-ray energies necessary to probe the Au L$_3$-edge transition has only recently become available. Hence, it is essential to validate the XANES spectrum of the unpumped sample to confirm that it is truly representative of the Au NPs. Figure S4 shows the steady-state XANES spectra of Au foil and nanoparticles measured at the Au L$_3$-edge transition ($2p_{3/2} \rightarrow 5d$) at the synchrotron. Au has a [Xe] $4f^{14} 5d^{10} 6s^1$ electronic structure, i.e., with a filled $d$-shell, which results in a weak absorption edge only visible due to some level of $s$-$d$ shell hybridization. For comparison purposes, the signal was plotted against Pt ([Xe] $4f^{14} 5d^9 6s^1$) (Fig. S5), illustrating that this method is sensitive to empty states within the metal $5d$ shell and, to some extent, the $s$-shell due to this hybridization.

The unexcited XANES spectrum of the Au NPs, measured at XFEL and the synchrotron, displayed a consistent shape (Fig. S4). The consistency between the synchrotron and XFEL data validates that the XANES acquired at XFEL accurately reflect the electronic structure of Au atoms. Additionally, incorporating time-resolved measurements into XAS enhances the method's ability to capture transient alterations in the electronic structure of gold prior to any detectable sample damage, i.e., probe-before destruction concept[58,59]. Moreover, the unexcited XANES of the Au NPs used were identical to the bulk gold spectrum, consistent with previous reports[23,60,61]. The observation that the electronic structure of the Au NPs used in this study matches that of bulk Au was further supported by theoretical calculations of the Au DOS as a function of particle size. These calculations indicate that for particles above 3 nm, the electronic structure of Au NPs begins to resemble that of bulk gold (Figure S6). Consequently, the similarity in the XAS spectral shapes observed for the Au NPs and bulk Au foil (Fig. S4) is consistent with these findings.

Ultrafast time-resolved XANES data were acquired with the XFEL source as a probe, following the excitation of a $5 \times 10^{-3}$ M Au NP aqueous solution at 532 nm (~ 2.33 eV) (i.e., slightly to the red of the LSPR maximum), utilising a 15 nm full width at half maximum (FHWM) bandwidth, a pulse duration of ~75 fs, and a power density of 4 μJ within the $60 \times 60$ μm$^2$ spot used for all time-resolved XANES experiments. The choice of this precise plasmon excitation energy was to induce LSPR excitation while minimising interband excitation[62]. The center of the Au $d$-shell is located at 2.5–2.58 eV (~496–480 nm) from the metal $E_F$[63,64], meaning that the laser pulse with a $2.33 \pm 0.13$ eV (15 nm FHWM) photon energy can only excite the low-energy tail of the $d$-shell at best. Therefore, the optical photon energy reduces to a high degree the interband excitations with an absorption onset at 2.38 eV[65] if one

ensures that only fundamental dipole LSPR transitions of Au NPs are excited[66], as controlled by laser fluence.

Since completely avoiding interband excitation when exciting close to the LSPR maximum is not feasible[62], it was crucial to demonstrate that its contribution to the overall signal remains minimal when exciting to the red of the LSPR peak[67]. To investigate this, optical TAS measurements were performed at different excitation wavelengths: 450 nm (predominantly interband excitation, below the LSPR peak), 520 nm (resonance excitation, at the LSPR peak maximum), and 532 nm (intraband excitation, above the LSPR peak). A comparison of the kinetic traces extracted near the excitation wavelength (Fig. S7) reveals that pure interband transitions (450 nm) are significantly less efficient than intraband transitions (532 nm) in generating hot carriers. Additionally, the observed decrease in signal amplitude when exciting beyond the LSPR peak, compared to excitation at the LSPR maximum, further supports the conclusion that interband excitation has a diminished contribution when excitation wavelengths are chosen to the red of the LSPR maximum[67].

Figure 2D compares the XANES spectra of unexcited (unpumped spectrum) and excited (pumped spectrum) recorded at $\Delta t = 100$ fs time delay after excitation at 532 nm. Optical excitation induced a spectral broadening, with the low-energy edge becoming more extended and the white line weakening, corroborating the presence of light-induced changes in the gold electronic structure around its Fermi-level energy and confirming that TR-XAS can track these changes. To better illustrate these results, the XANES difference spectrum (pumped-unpumped XANES spectra) is shown in Fig. 2D. The difference spectrum is dominated by the positive signal below and the negative signal above the Au $E_F$ (11,920.3 eV). Transient L$_3$-edge XANES clearly captures changes in state occupancy, particularly those induced in the $d$-shell, either directly or through processes like hybridization with the $s$-shell. Accordingly, a positive signal correlates with an increase in density of states (DOS); conversely, a negative signal (i.e., a bleached signal) indicates a decrease in empty states. Thus, the positive signal observed below the Au $E_F$ is attributed to the formation of a hot hole population from the non-radiative decay of the plasmon. Conversely, the negative signal observed above the Au $E_F$ is due to hot electrons filling empty states, as expected.

To exclude additional effects during the time-resolved XAS experiment, we performed additional test measurements at two different X-ray fluxes, with a two-fold increase in flux. The average X-ray flux at the sample position at 11,900 eV (monochromatic beam) was about $5 \times 10^9$ photons/pulse, corresponding to $2.5 \times 10^{13}$ W/cm$^2$ at applied experimental conditions, i.e., 75 fs X-ray pulse length and $60 \times 60$ μm$^2$ spot size. The transient XAS spectra measured at 100 fs time delay and two different X-ray fluxes equal to c.a. $3.7 \times 10^9$ and $7.5 \times 10^9$ photons/pulse, respectively, are plotted in Fig. S8. No detectable differences are observed between the spectra, indicating the absence of any observable nonlinear or multiphoton X-ray interactions. To induce any nonlinear interactions in the hard X-ray regime, fluences in the $10^{18}$–$10^{20}$ W/cm$^2$ range would be required, as reported in the literature[68–71]. Therefore, our experimental tests align well with previously reported studies, indicating that our specific experimental conditions were orders of magnitude lower than those reported to be required for multiphoton ionization or nonlinear processes.

An important consideration in the TR-XAS experiments is the pump laser fluence. Figure S9 presents the fluence dependence of the TR-XAS signal at an incident energy of 11,916 eV with a pump-probe delay of 100 fs. While TR-XAS ΔA at 100 fs shows an increase with rising laser fluence, the rate of this increase is relatively modest (slope = $11.1 \pm 3.5$), particularly in comparison to the TAS ΔA versus laser fluence slope $383 \pm 95$ (see Fig. S10). This indicates that the observed rise in TR-XAS ΔA does not significantly alter the overall electron temperature. Consequently, the TR-XAS signals remain within the ~2000 K electron temperature plateau, achieved at ~5 mJ/cm$^2$ of

laser excitation. Thus, to balance optimal signal intensity with minimal saturation effects, a fluence of 98 mJ/cm² was selected for the TR-XAS measurements.

The TR-XAS signal directly indicates hot carrier generation via LSPR non-radiative decay (Fig. 2D). The chosen laser pump energy minimizes interband excitation, ruling out direct vertical interband transitions as a significant decay pathway. Furthermore, the carriers' average energy distribution is larger than $\hbar\omega/4$ ($\hbar\omega$ = photon energy), excluding EE Umklapp scattering-assisted transitions. Therefore, one is left with phonon (or defect) scattering or diagonal transitions caused by Landau damping or surface collisions as potential electron decay channels. Notably, the hot hole and electron signals are neither symmetric nor have the same integrated magnitude. The carriers' wider and asymmetric distribution suggests that Landau damping[72] is the dominant electron decay process in smaller absolute space confinement, i.e., small nanoparticles[21,22,56,73]. This conclusion is further substantiated by a) the high crystalline quality of the Au colloids (see HRTEM Fig. 1B), suggesting a very low number of crystal defects present to induce scattering events, and b) the average particle size (ca. 7 nm) is significantly larger than the quantum limit of the plasmon (ca. 2 nm) and the intermediate regime (ca. 4 nm)[74], considerably reducing the rate of surface collisions. The signal asymmetry and finite-size effects are discussed further below.

To establish the time scales for plasmon Landau damping and the average lifetime of carriers, kinetic traces were extracted at the maximum of the hot hole intensity (11,916 eV, 4.3 eV below Au $E_F$) and the excited electron intensity (11,922 eV, 1.7 eV above Au $E_F$ populations, as depicted in (Fig. 3A). The kinetic data (Fig. 3C) from the time scans were fitted by a model published elsewhere[75] and described in the SI by equations S2 and S3. In brief, the data collected at 11,916 and 11922 eV were fitted with a convolution of a temporal instrument response function (Gaussian) with a monoexponential decay. The resulting fit is shown as the solid green curve in Fig. 3C. Due to the low signal-to-noise ratio for the hot electron data, the error bars are relatively large. However, it is possible to appreciate that the signal has dynamics similar to the hot holes. Note that the position of the Fermi level did not change significantly over the measured time scale (Fig. 3B), corroborating that the carrier transient changes are due to their relative populations.

Rossi et al. divided the total energy stored in the excited electronic system into the energy of nonresonant electron-hole transition contributions constituting screened plasmon excitation occurring in <10 fs and resonant transition contributions, comprising mainly hot carriers with a 17 fs lifetime (a.k.a Landau damping)[21]. The dephasing time of nonresonant electron-hole transition contributions is determined by analysis of changes in the spectral line shape caused by surface plasmon resonance. The induced broadening ($\Gamma_{hom}$, being the homogeneous linewidth of the surface plasmons resonance) can be determined by Lorentzian line fitting, and the dephasing time ($T_2$) is estimated from $T_2 = 2\hbar/\Gamma_{hom}$[44], with $\hbar$ being Planck's constant. Dephasing experiments on single particles using scanning near-field optical microscopes or nonlinear photoemission electron microscopy estimate the dephasing times between 5–9 fs[65,76], consistent with the calculations of Rossi et al.[21]. The plasmon Landau damping time can be extracted from the onset of XAS spectral shape changes caused by hot carrier formation, which start at ~24.6 ± 10 fs after optical excitation. This value is close to the previously calculated value[21]. Landau damping time and its associated error were estimated from the average of the onset of the rising function fitting (i.e., time zero) hot electrons and hot holes.

Following plasmon Landau damping, the hot carriers reach a maximum carrier population at 105 ± 8 fs, estimated from the rising edge analysis performed for hot electrons and hot holes. This value is consistent with optical measurements that revealed that the initial electron-electron scattering convolved with the rise of the transient absorption signal occurs within ~110 fs, corresponding to the average value of the five measured fluences[77]. The initial electron-electron scattering occurs after the Landau damping process, with the maximum populations expected to occur shortly thereafter.

The lifetimes of the hot carriers were determined from a single exponential decay to be 498 ± 35 fs and 505 ± 65 fs for hot holes and electrons, respectively. The ability to fit the data with a single exponential decay further supports the argument that interband excitations are largely avoided, as these excitations exhibit distinct kinetic decays (see Fig. S11). If interband excitations contributed significantly to the signal, a more complex decay behavior would be expected. Note that, to improve the fit for the hot electron data, which has a significantly lower signal-to-noise ratio compared to the hot hole data, the fitting was focused on the decay portion of the process. The rising parameters, derived from the hole kinetic data, were applied to aid in fitting the electron trace, as these parameters should be consistent across both datasets. The complete electronic thermalisation occurring within ~1.5 ps, consistent with the $\tau_{e-ph}$ of about 4–5 ps established with TAS, confirmed the ultrafast hot carrier relaxation as the primary bottleneck limiting plasmonic applications. The estimated thermalization time for hot carriers is consistent with previous studies where LSPR transitions are excited without significant interband excitation[78] and the predictions of the classic two-temperature model[49].

A recent study reported slow relaxation kinetics in TAS measurements alongside a predominant fast decay within 4 ps[79], which accounts for most of the observed signal. This slower, minor component was extracted through deconvolution of the optical signal under conditions where both interband and intraband transitions were excited. However, optical deconvolution offers limited ability to separate contributions from hot carriers versus phonons and does not provide direct insights into carrier energy. This limitation is particularly relevant as the measurements were conducted on larger Au nanoparticles (approximately 25 nm), which increases the likelihood of multipole excitation, potentially leading to longer-lived, lower-energy carriers[80]. Our TAS measurements also show a small, longer-lived component (Fig. S10), especially when exciting at the LSPR maximum, which we primarily attribute to phonon-phonon scattering[81] rather than an additional electron-phonon contribution. Given that the energy resolution of our transient XANES does not respond to phonon modes and that the signal fully decays within 1.5 ps, we conclude that the reported longer-lived signal is most likely attributable to phonons rather than hot carriers.

To estimate the number of electrons engaged when exciting 5 mM Au NPs at 532 nm, utilising a 15 nm full width at half maximum (FHWM) bandwidth, a pulse duration of approximately 75 fs, and a power density of 98 mJ/cm², the positive signal variance at 0 and 100 fs was integrated. This integrated signal was then juxtaposed with the signal difference between the Au and Pt L₃-edges (Fig. S5). Note that the signal difference between Au and Pt relates to $1e^-$ less in Pt valence states, i.e., the integrated positive signal of the difference between Pt and Au corresponds to the equivalent of having $1e^-$ from each Au atom participating in the resonance. Employing this simple methodology, we estimate that each gold atom contributes with $0.19e^-$ at the start of the resonance, which undergoes multiplication until 105 fs, reaching a maximum of $0.46e^-$ from each Au atom contributing to hot carrier formation at this excitation power. An Au NP has ≈ 10,000 atoms, equating to about $1.4 \times 10^{12}$ Au atoms in the excited volume. The photon density in the optical pulses is about $10^{13}$, from which 20% is absorbed according to UV-Vis, implying that the excited volume absorbs around $2 \times 10^{12}$ photons. This suggests an excitation of about $1e^-$ per atom of Au, of which 19% are converted into hot carriers at the onset, multiplying to about 46% within 100 fs. The observation suggests that hot carrier generation is a prime decay channel of Au NP LSPR and undoubtedly the most significant mechanism in nonradiative decay.

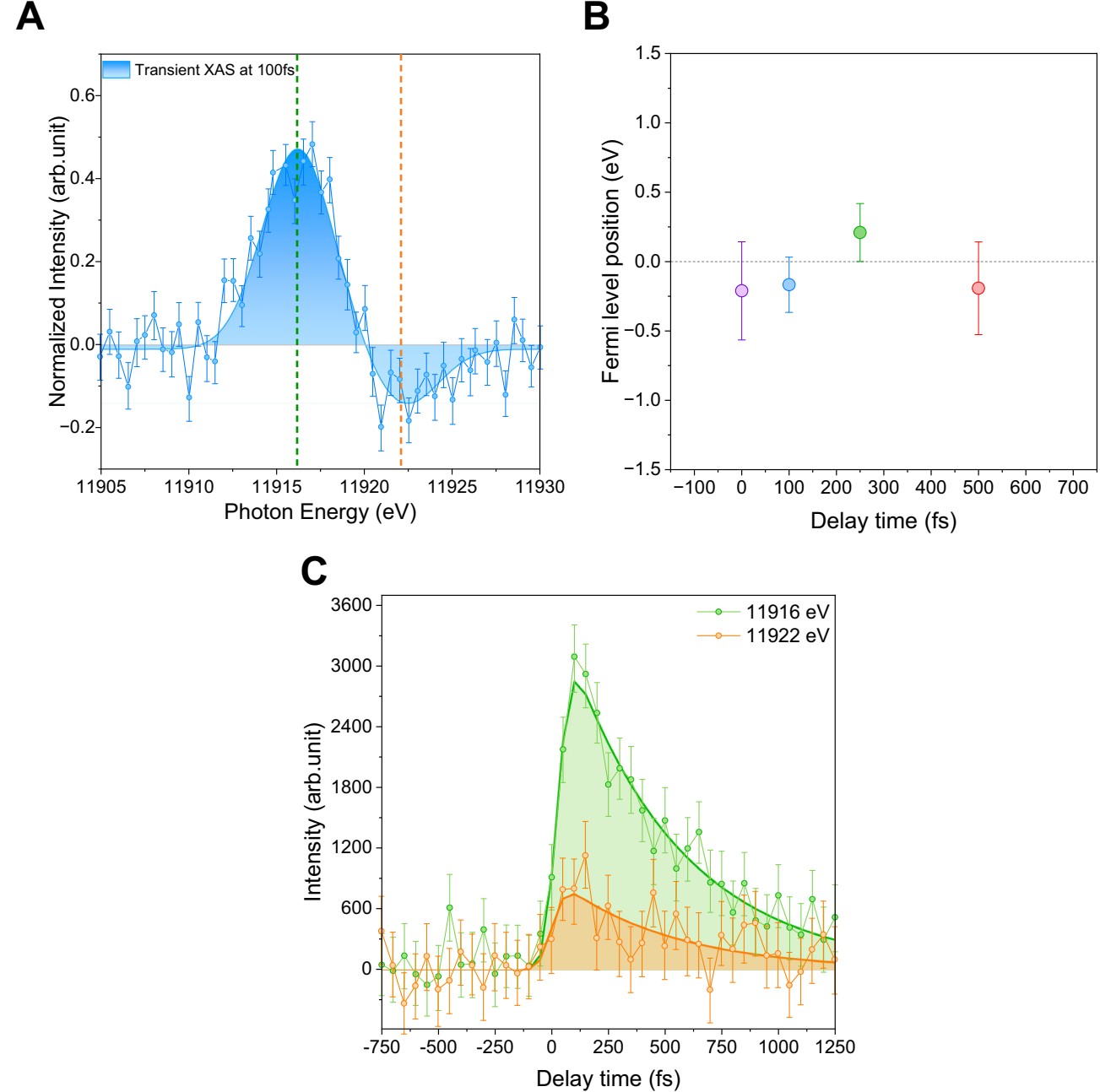

**Fig. 3 | Temporal evolution of the generated hot carriers. A** The difference spectrum (pumped-unpumped signal) shows kinetic traces of energy extraction points. **B** Transient changes of the zero intercept with energy scale (Fermi level). **C** Time traces showing intensity versus time delay extracted at X-ray photon energies of 11,916 eV (representing hot holes, green trace) and 11,922 eV (representing hot electrons, orange trace). The solid line represents the fit obtained using the methodology detailed in ref. 75 and described in the SI. The signal of the hot electrons was inverted in order to be plotted on the same y-axis.

## Energy distribution of plasmonic hot carriers

After verifying the generation of hot carriers, we proceeded to investigate the dynamic behavior of the hot carrier energy distribution—a significant yet elusive aspect in the realm of plasmonic hot carriers, particularly when it comes to holes[32,33]. The current understanding is derived mainly from theoretical studies[20,82,83] and indirect techniques[23,32,33,84]. For example, internal quantum efficiency measurements have inherent limitations as they solely quantify carriers transferred to an acceptor layer, like semiconductors, not their energy. Moreover, internal quantum efficiency does not provide information about the carrier's dynamic behavior within the metal. In Au, hot electrons can only populate the empty states within the *sp*-shells, but the holes can be in *sp*- and *d*-shells, confirmed by valence band−X-ray

photoelectron spectroscopy (VB-XPS) shown in Fig. 4A. It is evident when the VB-XPS is overlapped with the transient XANES spectrum (recorded at time zero) that photogenerated holes are located throughout the entire valence band of the metal, including the *d*-shell, despite the optical pulse energy allowing primarily *sp*-shell excitation.

Figure 4B illustrates the temporal evolution of the hot carrier population and their energy distribution, resulting from the non-radiative decay of optically excited LSPR transitions. As expected, this non-radiative deexcitation of the plasmon depopulates states below the Fermi energy and populates states above it. The ultrafast carrier-carrier interaction during carrier multiplication determines their energy and respective population. The hot carrier energy distribution exceeds single photon energy for hot electrons and holes.

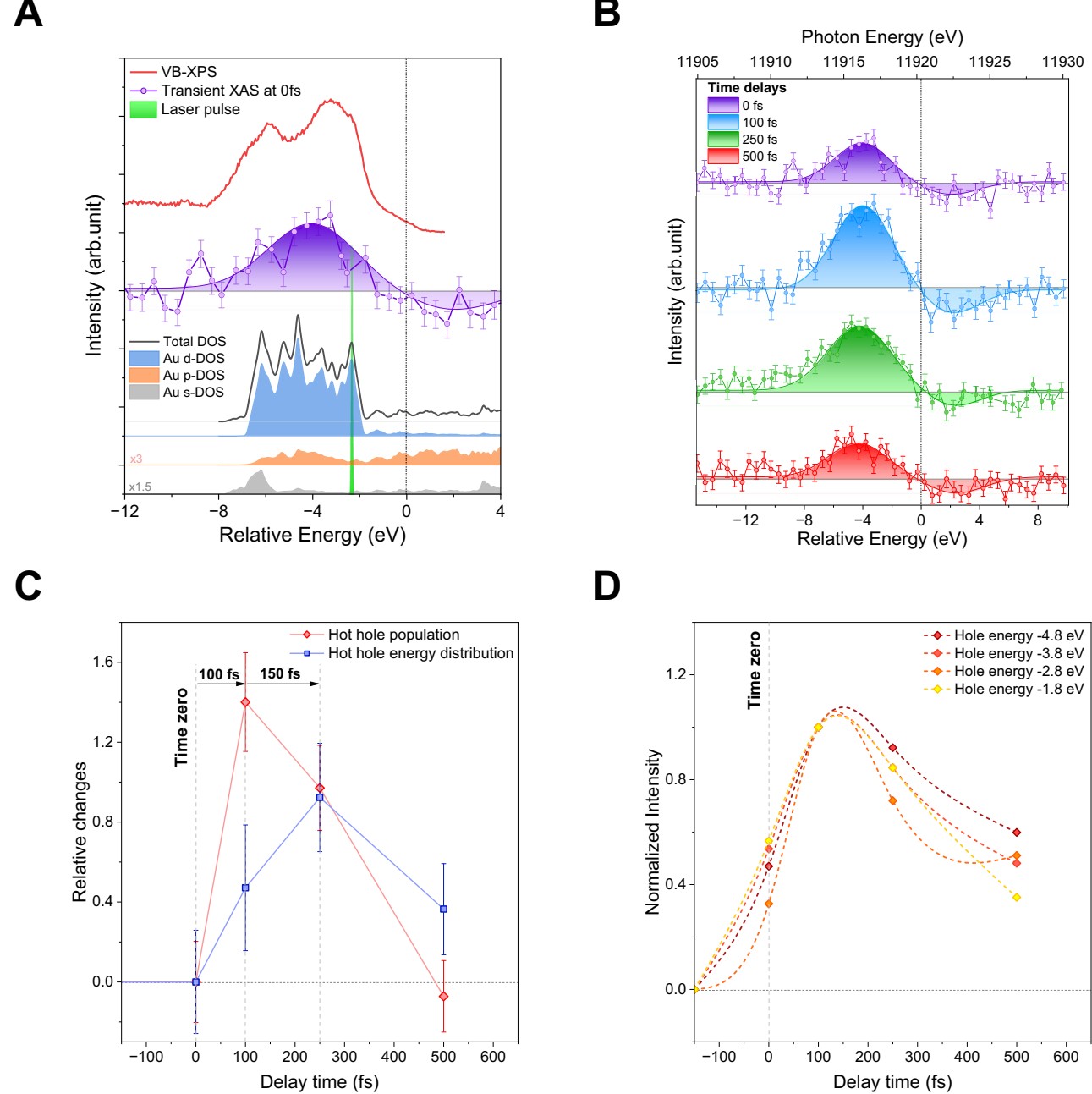

**Fig. 4 | Ultrafast energies distribution and kinetics of the excited state in gold nanoparticles. A** Comparison of the valence band photoelectron spectrum (VB-XPS) with the Au L$_3$-edge transient XANES spectrum collected at time zero. The zero of the energy scale corresponds to the Fermi level. The optical pulse in green is depicted as a Gaussian band centered at 2.33 eV with a full width at half maximum (FWHM) of 15 nm, corresponding to an energy envelope of ±0.13 eV; **B** The transient

XANES measured at the Au L$_3$-edge absorption spectra collected at different pump-probe time delays (0 fs corresponds to the best possible overlap between pump and probe). **C** Relative changes in hot holes mean energy distribution (blue trace) and population (red trace). **D** The temporal evolution of the hot holes with different energies indicates that the Fermi-Dirac distribution is not reached within 500 fs. Note that the energy of the holes is affected by the Au core-hole lifetime.

Furthermore, it is evident that the carrier populations and their energy distributions do not peak simultaneously. Additionally, an asymmetry is observed between the hot electron and hot hole populations.

The temporal evolution of hot carrier populations and their energy distribution following optical excitation can be understood without invoking finite-size effects. As shown in the DOS plots (Fig. S6), electron and hole occupations vary with particle size, with smaller nanoparticles (Au25) exhibiting more distinct peaks than larger ones (Fig. S12). This highlights that finite-size effects become significant only below the plasmon quantum limit (<2 nm)[74]. Using particles with an average diameter of 7 nm avoids finite-size effects, maintains dipole

resonance dominance, and ensures stability under the XFEL beam. Our findings thus apply to nanoparticles above the plasmon quantum limit and below the multipole resonance threshold.

The signal asymmetry between hot electrons and holes can be partly attributed to the higher sensitivity of the L$_3$-edge XANES transition to the formation of empty states in the *d*-shell, i.e., the hot holes. However, the shape asymmetry between hot electrons and holes is also anticipated because of the difference in electron and hole density of states, consistent with experimental[56] and theoretical reports[74]. Temporal analysis of the relative changes in mean energy distribution and population for hot holes (Fig. 4C) and electrons (Fig. S13) reveals a

contrasting behavior. Despite the significant experimental errors related to the lower sensitivity of the XANES probe to occupied states, the hot electron energy distribution and population have similar dynamics, peaking in intensity around 100 fs (Fig. S14). Electron-nuclear dynamics calculations within the Ehrenfest ansatz, implemented in the DFTB+ code[85], were performed to rationalise the observed asymmetry. Figure S12 shows an apparent asymmetry in the electron and hole dynamics, owing to the asymmetric DOS around the Fermi level within the range of the laser energy. These observations are in qualitative agreement with the experimental measurements. The higher localisation of electrons concerning the holes favors carrier multiplication, which increases the number of carriers and simultaneously reduces their energy[86].

The rapid depopulation of electrons in the $d$-shell is expected due to the overlap between the $d$ and sp-shells. Consequently, a high density of $d$-electrons will couple with the plasmonic resonance to dissipate its energy[83]. This is consistent with the time-resolved calculations of electron and hole localisations shown in Fig. S12. However, this does not explain the observation of carriers with energies above the photon energy, even considering the energy broadening induced by the 5.41 eV Au $L_3$-edge core-hole broadening[87], which limits the experimental energy resolution[88]. Nonetheless, hot holes are distributed across the entire valence electronic structure, and their energy distribution increases up to 250 fs (Fig. 4C) before starting their relaxation. These two observations imply the involvement of carrier-carrier coupling mechanisms that both increase carrier population and its energy distribution, an effect that has yet to be reported[76]. Note that the experimental conditions preclude the possibility of multiphoton excitation of single electrons.

When it comes to carrier multiplication, there are two possible scattering mechanisms: impact excitation and Auger heating[89,90]. The predominant mechanism in carrier multiplication is impact excitation. In the impact excitation mechanism, an excited electron (hole) undergoes Coulomb scattering, losing energy and momentum and giving rise to an additional electron-hole pair. The distinctive feature of impact excitation is a rise in the number of carriers and a simultaneous reduction in their energy. Conversely, Auger heating characterizes the non-radiative recombination of an electron with a hole, where the energy and momentum are transferred to an electron (hole) within the same shell. The hallmark of Auger heating is a decline in the number of carriers and an increase in their energy.

To enhance the visualisation and comprehension of the hot hole multiplication process, the integrated hole population and the energy distribution (aka energy width ($3\sigma$)) are plotted versus delay time (see Fig. 4C) using the data analysis procedure outlined in the SI. Commencing with the average hot hole population, it peaked at 100 fs and decreased subsequently. This implies the maximum nonequilibrium non-Fermi-Dirac hot carrier population occurs after Landau damping (early hot carrier population), indicating the involvement of the impact excitation scattering mechanism. When examining the hot carrier energy distribution width, it is noticeable that it increases up to 250 fs. This observation indicates the involvement of Auger heating in the carrier multiplication, but, more importantly, the mechanism involvement extends beyond the 10 s of fs[90], hugely significant for hot carrier applications.

A final aspect of plasmon carrier relaxation that can be observed is the time when the Fermi-Dirac distribution is reached. For extended metal surfaces (e.g., Au thin films), time-resolved studies suggest forming a Fermi-Dirac-like distribution characterized by a sizeable effective electron temperature within 1 ps[56,91]. However, in Au NPs, the electron gas is expected to thermalise very fast to a Fermi-Dirac distribution over a time scale $\tau_e$ -100 fs[92,93]. In such a scenario, one would expect carriers with different energies to decay at different rates, with the ones with the highest energies decaying more rapidly. Figure 4D (for hot holes) and S14 (for hot electrons) illustrate carrier decay across different energy

levels over a 500-fs period. Interestingly, the decay rates are similar regardless of carrier energy, suggesting that achieving a Fermi-Dirac-like distribution with a high effective electron temperature takes longer than the expected 100 fs. To determine the precise time scale for this thermalization, time-resolved, high-resolution XAS and XES would be required, which were not accessible during this measurement.

The participation of the Auger heating mechanism in carrier multiplication helps explain early reports concerning hydrated electron formation with Cu NPs[94] and near-infrared plasmon-assisted water oxidation, which used photons with insufficient energy to drive such processes[95]. More critical is the finding that the Auger mechanism extends up to 250 fs and the nonequilibrium Fermi-Dirac distribution extends beyond 500 fs, meaning that the hottest carriers are available and able to perform work, according to the Franck-Condon principle and Marcus theory. This is transformational for photocatalysis, photovoltaics, solar redox flow batteries, and phototherapy with hydrated electrons because it enables low-energy photons to do work on applications requiring voltages beyond the ones attained with single photon energy. In photocatalysis, one can foresee driving chemical reactions with redox windows that are more extensive than the photon energy, avoiding the use of detrimental high-energy photons. Similarly, one can generate hydrated electrons in situ with lower-energy photons with a higher penetration depth. In photovoltaics, one could create devices with larger open circuit voltages than a single photon permits, enabling effective photon energy use and quite possibly circumventing the Shockley-Queisser limit for single junction solar cells.

As a final remark, it is important to reiterate that while the excitation wavelength used for transient XANES may induce a small fraction of interband transitions due to plasmonic near-field enhancement, our experimental design was carefully chosen to minimize this effect. By selecting an excitation wavelength slightly red-shifted from the LSPR maximum[62] and positioned relative to the expected Au $d$-band onset[63,64], we effectively reduce the likelihood of significant interband contributions. The validity of this approach is further supported by the similar dynamics reported by Sun et al.[78], who utilized 900 nm excitation to completely suppress interband transitions. Therefore, the collective evidence strongly indicates that interband transitions do not play a dominant role in the observed signal at 532 nm, even when accounting for potential near-field enhancements.

In this work, the dynamic behavior of plasmon hot carrier formation, multiplication and thermalisation on gold nanoparticles upon LSPR excitation is reported. The methodology is intrinsically sensitive to the metal electronic structure, permitting real-time observation of the entire process. The plasmon Landau damping was determined to be $\sim 25$ fs, with a maximum hot carrier population detected at 105 fs after excitation. At this time point, there is $\sim 0.46e^-$ per Au atom as a hot carrier, establishing hot carrier formation as a significant decay pathway of plasmon excitation. Complete thermalisation of the hot carriers occurs $\sim 1.5$ ps. Energy scans at variable delay times reveal that carriers do not reach a Fermi-Dirac distribution within 500 fs, signalling that the high-energy carriers can be realistically harnessed to do work. Importantly, carriers with energies exceeding the single photon excitation energy were detected, suggesting the involvement of the Auger heating scattering mechanism in the carrier multiplication apart from the expected impact excitation mechanism. This observation opens perspectives for plasmon hot carrier applications in fields where the carrier energy defines the device's potential for work, such as photocatalysis, phototherapy with hydrated electrons and photovoltaics. These insights into plasmon-induced hot carrier generation and dynamics provided here further the fundamental understanding of plasmon hot carriers and are likely to impact applications for years.

## Methods
All details regarding the experimental methods are provided in the Supplementary Information.

## Data availability

All data needed to evaluate the conclusions in the paper are present in the paper/Supplementary Information/Source Data file. Source data are provided with this paper. Additional data supporting this study's findings are available from the corresponding author upon request. Source data are provided with this paper.

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

## Acknowledgements

We acknowledge the Paul Scherrer Institut, Villigen, Switzerland, for providing beamtime at the Alvra beamline of the SwissFEL facility. We also acknowledge SOLARIS National Synchrotron Radiation Center, Krakow, Poland, for the access to the ASTRA and PHELIX beamline. The simulations were performed using computational resources provided by the Swedish National Infrastructure for Computing (SNIC) at UPPMAX and NSC, for which we want to thank. Special thanks to Joanna Kowalik from NCPS SOLARIS for her valuable assistance with the illustrations (Figs. 1A, 2A). Funding Olle Engkvists Stiftelse grant 210-0007 (J.Sa); Knut & Alice Wallenberg Foundation grant 2019-0071 (J.Sa); Swedish Research Council grant 2019-03597 (J.Sa); European Union's Horizon 2020 research and innovation program under Marie Skłodowska-Curie grant 884104 (PSI-FELLOW-III-3i) (A.W.); Robert A. Welch Foundation grants C-1220 (N.J.H.) and C-1222 (P.N.); Air Force Office of Scientific Research via the Department of Defense Multidisciplinary University Research Initiative under AFOSR grant FA9550-15-1-0022 (N.J.H., P.N.); National Science Center in Poland grant 2020/37/B/ST3/00555 (J.Sz.); Polish Ministry and Higher Education project: "Support for research and development with the use of research infrastructure of the National Synchrotron Radiation Center SOLARIS" grant 1/SOL/2021/2.

## Author contributions

Conceptualization and methodology: A.W., J.Sz. and J.Sa.; formal data analysis: A.W., J.Sz. and J.Sa.; experimental investigations: A.W, R.B.-V., C.B., C.C., P.J.M.J., R.G.C., V.R.S., P-B., J.K., A.M., T.S., E.P.-J. and J.Sa.; data visualisation concepts: A.W., J.Sa., J.Sz., and N.J.H., draft preparation: A.W., N.J.H., P.N., J.Sz. and J.Sa.; writing-review and editing: all the authors. All authors have read and agreed to the published version of the manuscript.

## Funding

## Competing interests

The authors declare no competing interests.
