## [Transparent Peer Review file · Nature Communications]

The Dynamics of Plasmon-Induced Hot Carrier Creation in Colloidal Gold

Corresponding Author: Professor Jacinto Sa

Version 1:

Reviewer comments:

Reviewer #1

(Remarks to the Author)

This article presents an ultrafast X-ray absorption study of colloidal Au nanoparticles excited via their plasmon resonance. The use of X-ray absorption is ideal here as it allows a detailed visualization of the electronic structure changes incurred by the system after photoexcitation. The main conclusions are a commensurate hole and electron relaxation, a finite risetime for the population of charge carriers, which the authors interestingly attribute to the occurrence of Auger processes. The paper deserves publication but it contains a number of statements, which the authors ought to modify:

- bottom of p. 2 and top of p. 3, the authors introduce XAS, but the last 2 sentences of this § speak about optical excitation, they then jump back to XAS in the following §.
 - p. 3, 2nd §: the authors mention a time resolution of 5 ps for synchrotrons. While this is achievable, it is rather uncommon. The temporal resolution is more around 50-100 ps.
 - p. 7, 1st §: the authors conclude their discussion on the Temperatures, started the page before, but they do not conclude so a logical link can be made with the following §.
 - p. 7, 3rd §: The statement "Au NPs have yet to be measured at XFELs" is to the reviewer quite shocking since Au NPs have been among the most studied systems at XFELs. Ultrafast X-ray diffraction, coherent diffractive imaging, X-ray photon correlation spectroscopy, X-ray holography, etc. have been performed at XFELs using Au NPs, which serve almost as the ideal test model.
 - Same §, what is a "slight absorption"? You mean a weak absorption?
 - p. 9, 2nd §: The analogy between the bulk and NP electronic structure for sizes above 4 nm has long been established (see Doremus, J. Chem. Phys. 1964, 40, 2389–2396; Kawabata et al., J. Phys. Soc. Jpn. 1966, 21, 1765–1772). It is sufficient to say that the present results fit into this trend.
 - p. 10: I would move §§ 2 and 3 to the SI. These are checks of the fluence dependence that are not directly relevant to the conclusions of the paper.
 - Fig. 3: I would overlap the traces in panels C and D after flipping the latter. This will better visualize the correspondance between the 2 traces.
 - p. 13: Given the very noisy electron trace (fig. 3D) I am surprised that the uncertainty of the fit is only 65 fs, barely twice that of the hole trace.
- A couple of minor points:
- Be careful at the confusion between kinetics and dynamics. The former relates to population decay, the latter to wave packets. Here the authors deal only with kinetics, but they use "dynamics" in page 13 and fig. 4.

Reviewer #2

(Remarks to the Author)

After carefully reviewing this manuscript, unfortunately I have to conclude that I cannot recommend its publication in its current form. I agree with all the other reviewers that the data are very unique and potentially very interesting. However, in my opinion the study is fundamentally flawed as it does not have any control experiments for the ultrafast measurements. Naturally this leads to many questions that were raised by previous reviewers.

I will highlight only one example here: 532 nm excitation CAN excite interband carriers in gold since bulk gold does absorb in this region. Indeed this has been confirmed in ultrafast measurements as well in many of the papers cited by the authors. The authors might be correct that this is not the main contribution in their experiments, however no logical arguments or back-of-the-envelope estimates can convince me in this unless there is a control experiment on e.g. bulk gold, thin films etc. Without such measurements, the entire argument regarding plasmon dephasing, which is the main premise of the paper, collapses.

Similarly, other significant claims in the paper are open to alternative interpretations without control experiments. As one of the reviewers mentioned, we all understand that FEL experiments are challenging, and we appreciate the novelty of applying these measurements to Landau damping in plasmonic structures. However, we must uphold basic standards of scientific rigor.

Reviewer #3

(Remarks to the Author)

This manuscript describes a high-level experimental effort to apply ultrafast time-resolved X-ray absorption experiments to help determine the dynamics and energy levels of “hot” holes and electrons in plasmonic gold nanoparticles. The topic is well-introduced and the manuscript overall is well-written. I think the work will have strong interest from Nature Communications readers. The authors report hot carrier energies that extend well beyond the more typical limit of the pump photon energy, which will garner significant interest, and may be well justified since this is a new experimental approach to monitor hot carriers. I do have a few small suggestions that if addressed could increase impact:

1. Figure 1a could probably be optimized a bit further. It shows e-ph dynamics extending to 10ns, but I believe the authors may have intended to ph-ph dynamics to the text in figure? It might just be a small omission as 10 nanoseconds is too long for e-ph interactions. The authors say themselves in the text that e-ph interactions occur on a “picosecond time frame.” They also say in the text on page 13 that ph-ph relaxation occurs in “10s of nanoseconds.” I think Figure 1a could just use a little optimization to be more consistent with the text.
2. I am a little confused why the authors say (line 116) that the pump beam in transient absorption is “usually” in the infrared range? Isn't it usually just chosen to excite plasmon, and for Au nanoparticles that are spherical, that is in the visible? In the work in this paper, the excitation is in the visible at 535nm. I might be missing the point that the authors are trying to make here.
3. It is best practice in transient absorption experiments to show the kinetics plots extend long enough to show the signal go to zero which helps assure the reader that there is reduced likelihood of artifacts (such as the presence of a long-time signal that is re-excited by the next pulse). If the authors have the longer time data for Figure 3, it would be nice to plot. I recognize that these are hard experiments, so if they are not available it is not absolutely necessary.
4. Similarly, in Figure 4b, why not show spectra extending to longer times? Figure 4b goes to 500fs, but Figure 3 shows kinetics out to 1.25ps. Maybe the additional spectra could be put in the supplemental information, since too many spectra could take away from a very nice Figure 4.

Reviewer #4

(Remarks to the Author)

This paper reports very interesting FEL-XAS experiments on carrier dynamics in Au nanoparticles following plasmon excitation. Sample characterization by in-house methods is sound. The discussion is aided by ab-initio simulations. The FEL-XAS experiments per se were expertly conducted. The discussion contains interesting ideas and the paper, if published, will be of interest to researchers working in a very active area.

However, I cannot recommend publication in its present form because the discussion and interpretation suffer from over interpretation in the part related to carrier dynamics (error bars too large, data points too small) and because there are very numerous ambiguities and cases of sloppy presentation. A detailed list is found below.

It will be necessary to re-write the abstract, taking into account the comments below on the paper.

Major

- Line 100, Fig. 1D. The fits are not visible in Fig. 1D. You state that they are “single exponential”, but it seems the time dependence is more complex. See also later comment on lines 164 – 173.
- Line 104. Fig. 1A does not seem correspond to the physical phenomena described in this section. It would be better to depict separately plasmon excitation, followed by electron – hole generation due to plasmon de-excitation and subsequent damping and thermal dissipation. Also, the description of the phenomena in lines 104 – 114 could be more streamlined and clearer. Check also the comment on line 252.
- Line 164 – 167. You state that the increase in electron temperature was found by “the first exponential fit”. What does this mean? Also, the table in fig. 2B reports two relaxation times: e -ph and ph – ph. These are never described in the text. The equation used to fit the kinetic traces in Fig. 1D must be reported, indicating the fitting parameters.
- Lines 168 – 173. You state that the increase in electron temperature “stagnates”. Do you mean “saturates”? Any way, saturation is observed in Fig. 2C only for laser fluencies used in the TR-XAS experiments.
- Line 123. You state that the LSPR is centered “nominally” at 520 nm. Why “nominally”? In Fig. S3 the peak of the resonance appears to be at 550 – 560 nm. Is this the effect of the sloping background? Can you be more accurate in the determination of the energy of the LSPR? This would help in justifying the choice of 532 nm for the pump probe in the XAS measurements (line 230).
- Line 225 and Fig. S6. The NP size for all intermediate cases should be reported. It is surprising that the DOS curves do not

change gradually. The one in the middle is highly structured. Check.

- Line 248. The changes observed are presumably due to changes in the occupation of states not to modification of the density of states. These are different concepts.
- Line 252. The formation of holes is presumably due to the de-excitation of the plasmon following its optical excitation. The present wording is confusing. This is linked to the ambiguous phrasing in lines 104 -114 and Fig. 1A.
- Lines 268 – 274, Fig. S8 and its caption. This is very unclear. What is the relation between the plot of $\Delta(A)$ vs. fluence values and the electron temperature? How was the conclusion that the electron temperature is constant at about 2000 K reached (Fig. 2 C and caption to Fig. 8)? What is the meaning of the straight line fit in fig. 8? I have serious reservations on fitting the highly scattered points (no error bars) with a straight line, any function can pass through those points. Finally, it is stated that at 98 mJ/cm² there is no saturation, but the data in Fig. 2C seem to contradict this.
- Lines 275 – 277. There is an apparent contradiction. First, “generation of hot carriers through LSPR decay”. Then “the choice of pump energy excludes direct vertical transitions as a significant decay channel”. Rephrase, otherwise it is incomprehensible.
- Line 336. You state that the “thermalization time aligns with the measured value when exciting solely the LSPR transition”. What does this mean? Is there another excitation besides LSPR?
- Line 380. Again, there seems to be confusion in the use of “excitation” and “de-excitation”. It is the de-excitation of plasmons which leads to the variation of states above and below the Fermi level.
- Line 384, Fig. 4C. There are 3 experimental points for “population” and “distribution”. How were these determined, what is the meaning? Anyway, the error bars are very big, I do not believe it can be stated that they reach a maximum at different times, there is too much scattering of the data points; I think this is an over interpretation of the data.
- Line 388, Fig. 4A. It is misleading to indicate the photon energy as a single vertical line. The photon induces a transition, I suggest a horizontal arrow.
- Line 394, Fig. 4D. What is the meaning of the curves? The ones for – 2.8 eV have a surprising ondulation! Why is the energy of the holes “affected” by the Au core – hole lifetime? Do you refer to broadening? This is different.
- Lines 398 – 406. This paragraph and Fig. S9 are unnecessary. The particles used are 7 nm with small dispersion, there are no size effects. The same message has already been given in Fig. S6, why the repetition? In the caption: projected on what? Why is the photon energy reported as a vertical line? A photon induces a transition, it should be represented as a horizontal line, as already mentioned.
- Line 407 onwards. As already mentioned in line 384, I find all this discussion on the width and population of electrons and holes highly speculative and an over interpretation of the data. Error bars are too big and the number of points too small to be credible. At most, a qualitative interpretation might be acceptable.
- Line 407: higher sensitivity to the formation of hot holes with respect to what? Hot electrons? Why? In line 408 why “however”? Is this an explanation of the asymmetry?
- Line 415 and Fig. S10. The DFTB+ simulations are interesting and apparently provide support to the interpretation. However, details on the simulations should be provided (in the SI). Simulations for various NP sizes are unnecessary (see also comment above on line 398). Intermediate times should be provided in the figure, not just the minimum and maximum. The ordinate scale for the biggest NP should be changed in order to make the figure intelligible. Rather than the number of atoms in the NP their size should be given, in order to make the presentation consistent. In line 425 the simulations are referred to as reporting “localization”. Why? What do you mean?
- Line 454. The discussion of hole and electron dynamics (Figs. 4C and S12) must be improved. In Fig. S12 the intensity for three energy values actually increases at 500 fs; however it is just one data point. Is it credible? What is the expected behavior for a Fermi Dirac distribution?

Minor/formatting/language

- Lines 34 – 35. Surface plasmons have been known for decades and are a textbook subject. They have not “emerged” (it is implied, recently). Rephrase.
- Line 36. Why “distant” radiation? Distant from what?
- Line 37. Instead of “profound” use “high” or “significant”.
- Line 38. Instead of “attributes” use “characteristics”.
- Line 59. Instead of “interplay” you probably mean “interaction”. Anyway, use of XAS to investigate radiation – matter interactions is besides the point in this context, why quote it? You are using XAS to probe the evolution of the electronic structure following plasmon excitation, focus on that.
- Line 62. Following x-ray absorption core electrons undergo an excitation and perform a transition, they do not “shift”.
- Lines 65 – 66. This phrase is completely out of context.
- Line 69. Carrier “participation” in what? In what process?
- Line 81. I suggest to quote some other FEL – XAS studies of functional materials, in order to better represent recent activity in this field. For example: Y. Uemura et al., *Angew. Chem.* 55, 1364 (2016). Y. Obara et al., *Struct. Dyn.* 4, 044033 (2017); Y. Uemura et al., *Chemical Communications* 53, 7314 (2017); Pelli Cresi et al., *Nano Lett.* 21, 1729–1734 (2021).
- Line 125. The UV-Vis spectra (add).
- Line 134. What are “typical” lase fluences? Typical of what?
- Line 143. Instead of “variance” I think you mean “difference”. The variance refers to a distribution or population, not to two quantities.
- Line 154. Check the signs in Eq. 1. I suspect there should be a + sign on the RHS of the second equation.
- Line 179. What do you mean in stating that XAS allows tracking hot carrier “energetics”? Maybe hot carrier energy distribution?
- Line 184. I presume that the optical pulse is not “delayed” with respect to the X-ray one, rather the opposite. The time interval between the optical pulse and the X-ray one is varied.
- Line 193. It is Fig. S4.

- Lines 196 – 199. The absorption edge (i.e. the discontinuity) of the Au L3 edge is clearly visible. It is the white line which is greatly damped (with respect to Pt) because of the d10 electronic structure of Au. It is well known that XAS at metal L3 edges is sensitive to empty states in the 5d shell, so use “illustrating” rather than “revealing”, it is hardly a new discovery.
- Lines 218 – 219. What do you mean by “genuinely describe the Au NPs used in electronic structure”? Used in electronic structure? A single FEL – XAS spectrum does not “uphold” the ability to capture transient electronic structure. Rephrase, write more clearly.
- Line 237. Instead of “largely excludes” use “excludes to a high degree” or similar.
- Line 243. Corroborating the presence of light induced
- Line 248. The changes observed are presumably due to changes in the occupation of states not to modification of the density of states. These are different concepts.
- Line 252. Instead of “forming” use “formation”.
- Line 307. Define the symbol Γ (hom). Homogeneous?

Version 2:

Reviewer comments:

Reviewer #1

(Remarks to the Author)

The authors have addressed my concerns. The paper can be published.

Reviewer #2

(Remarks to the Author)

My main criticism of this paper was that it is fundamentally flawed as the authors have not performed control experiments. As an example I pointed out that the pump-probe results at 520 nm can be easily due to the interband excitation in which case the whole argument of the authors about plasmon-assisted effect falls apart.

To address this issue the authors have presented transient measurements results for different pump wavelengths and claim that these results prove that the excitation at LSPR wavelength are not dominated by interband excitation. I strongly disagree with this assertion. The difference in the amplitude of the transient signals in Fig S7 are most likely due to plasmon near-field enhancement of interband absorption. The lifetime variation in Fig. S11 is most likely due to thermal effects (electron-phonon coupling depends on heat capacity which is a function of the temperature). The authors can easily verify this by performing power-dependent measurements at LSPR resonance and observe longer decay times at higher excitation power densities. Furthermore, they cite Phys. Rev. B 50, 15337-15348 (1994) to support their claim, however the pump wavelength in that work was in the infrared (~900 nm) and the observed decay was simply due to electron-phonon thermalization and not plasmon decaying.

Unfortunately, this revision has only strengthened my conviction that this work is not suitable for publication.

Reviewer #3

(Remarks to the Author)

The authors have done a nice job addressing my comments/concerns in the manuscript and in their responses. I have no further requests/questions. These are very difficult experiments on a beamline that has limited access. Overall, I believe the readership of Nature Communications will find this work very interesting.

Reviewer #4

(Remarks to the Author)

The authors have responded in detail and in a satisfactory manner on my comments to the first submitted version. They have revised the paper accordingly. Fine by me.

Version 3:

Reviewer comments:

Reviewer #1

(Remarks to the Author)

I feel the paper can now be published. It is true that the debate about the contribution of d-transitions vs plasmon excitation is sound and raises a number of questions. However, overall it is fair to assume that even if d-excitation is present, the bulk of the observations comes from plasmon excitation. In addition, the results in this paper will surely stir further studies that will look more carefully into this issue.

Reviewer #2

(Remarks to the Author)

I feel that, at this point, we are unfortunately starting to go in circles. As I mentioned in my initial review, the relevant control experiment would involve conducting the same measurements (visible pump, X-ray probe) on bulk gold samples. This approach would isolate the contribution of plasmon dephasing. The power-dependent measurements presented by the authors do not constitute proper control experiments for a simple reason: they do not address the primary issue of this experiment—the potential for interband absorption at 532 nm.

The authors' claims rest on the assumption that interband absorption in gold can be neglected at 532 nm. Based on my experience and the extensive literature on this subject that I am familiar with, this assumption is not accurate.

Unfortunately, I am unable to provide further constructive feedback on this paper. I understand that these measurements are very challenging, and there may be little the authors can do at this stage. I sympathize with their efforts; however, I cannot overlook the fact that the experimental design appears fundamentally flawed, in my opinion.

Reviewer #4

(Remarks to the Author)

The authors have done a good job of responding in detail to all comments, especially those of referee 2. In my view, the paper should be published and will contribute significantly to research in this topical field.

Answer to Reviewers' comments:

First round of revision:

Reviewer #1:

In the manuscript titled “The Dynamics of Plasmon-Induced Hot Carrier Creation in Colloidal Gold” by Wach et. al. presents a comprehensive experimental evidence to establish the underlying mechanisms for plasmon hot carrier generation, multiplication and thermalization upon local surface plasmon resonance (LSPR) excitations in gold nanoparticles. The authors employed state of the art transient X-ray absorption spectroscopy (TR-XAS) where a laser pump was used to excite the system and an X-ray probe was employed to trace the hot carrier dynamics.

The manuscript is interesting with some “detailed” results. Notably, the authors provide experimental evidence for decay of surface plasmons by Landau damping which is an unexplored nonradiative pathway for plasmonic decay. Although Landau damping has been predicted theoretically, this work with TR-XAS experiments, reports the first ever direct observation of the mechanism. They achieved high energy non-equilibrium electronic distributions for longer than expected times which could be harnessed for transforming light into electrical currents and achieving photodetectors which surpass traditional limits. These results are supported with estimates of the number of electrons contributing to hot carrier dynamics, and analysis on carrier multiplication mechanisms. Given the growing interest in plasmon-induced hot carriers in optoelectronics, the work is relevant and timely. Therefore, I recommend this work for further considerations. However, the manuscript in its current form requires improvement to address some conceptual and technical gaps. My recommendation for publication is contingent on addressing them. I highlight these in a point-by-point manner below.

Reply: We thank the Reviewer for recognising the work's worthiness and dedicating time to the revision. We also thank the Reviewer for the valuable comments, which we addressed below and in the manuscript. Thanks to the Reviewer's comments, the revised version is more precise and has a higher scholar level. We hope the provided clarifications and additions removed the Reviewer's initial reservations, and thus, we can get endorsement for publication.

In addition to Landau damping, super-collision is another significant (and sometimes dominant) pathway to generate electron-hole pairs via resonant non-vertical transitions. It could potentially be a competing mechanism, for example, see *Nanophotonics* 9, 453-471 (2020) or Ref. 22 in the main text. Given a cascade of mechanisms behind hot carrier generation, how do authors ensure that the reported mechanism is Landau damping and differentiate it from competing mechanisms such as super-collision?

Reply: We thank the Reviewer for the comment. According to Khurgin (*Nanophotonics* 9, 453-471 (2020)), four possible electron decay channels lead to the formation of hot carriers. The direct "vertical" interband transition can be automatically excluded because of the pump pulse energy. Also, the carriers' average energy distribution is more significant than $\hbar\omega/4$ ($\hbar\omega$ = photon energy), excluding EE Umklapp scattering-assisted transitions. From the average carrier energy distribution analysis, one is left with the phonon (or defect) scattering or diagonal transitions caused by Landau damping or surface collision as potential electron decay channels. HRTEM showed that the colloids have low defect density, so this channel can also be omitted, leaving Landau damping or surface collision as the most promising ones, as the Reviewer highlighted the most. The carriers' wider and asymmetric distribution indicates Landau damping, as reported in several publications (e.g. *Nanophotonics* 9, 453-471 (2020); *ACS Nano* 14, 9963-9971 (2020); *Nat. Commun.* 9, 1853 (2018)). Furthermore, the average particle size (ca. 7 nm) is significantly larger than the quantum limit of the plasmon (ca. 2 nm), considerably reducing the rate of surface collisions (*ACS Photonics* 4, 2759-2781 (2017)). For those reasons, we are confident that the prime electron decay channel is Landau damping.

In lines 179 and 180, in reference to figure 1D, the authors say “The transient signal directly demonstrates the generation of hot carriers through LSPR decoherence via Landau damping” and cite Ref. 22. However, neither figure 1D does not seem to provide the above mentioned information nor Ref. 22 reveals the aforementioned information. It would be better if the text is modified accordingly.

Reply: We thank the Reviewer for the comment. The comment was largely answered in the previous reply.

Action taken: we updated the manuscript text with the statement below and several references for support:

The transient signal directly demonstrates the generation of hot carriers through LSPR decoherence. Since interband excitation is avoided mainly with the choice of laser pump energy, this excludes direct vertical interband transition as a significant decay channel. Furthermore, the carriers' average energy distribution is larger than $\hbar\omega/4$ ($\hbar\omega =$ photon energy), excluding EE Umklapp scattering-assisted transitions. Therefore, one is left with the phonon (or defect) scattering or diagonal transitions caused by Landau damping or surface collision as potential electron decay channels. Notably, the hot hole and electron signals are neither symmetric nor have the same integrated magnitude. The carriers' wider and asymmetric distribution indicates Landau damping, which is the dominant electron decay channel process in smaller absolute space confinement, i.e., small nanoparticles. The conclusion is further substantiated by a) the high crystalline quality of the Au colloids (see HRTEM Fig. 1B), suggesting low crystal defects for scattering events to occur, and b) the average particle size (ca. 7 nm) is significantly larger than the quantum limit of the plasmon (ca. 2 nm) and the intermediate regime (ca. 4 nm), considerably reducing the rate of surface collisions.

The hot carrier dynamics and population for holes and electrons exhibit asymmetry. In the text around figure 3B, the authors say that this cannot be explored fully due to the probe's lower sensitivity to hot electrons. While it is fundamentally true that XAS is more sensitive to the final unoccupied core state, the asymmetry is consistent and significant in all figures. Moreover, it aligns with the density of states in figure 3A. It would be better if this could be addressed at least at a qualitative or logical level, possibly by pointing out relevant transitions, using density of states as reference, or through shell hybridization.

Reply: We thank the Reviewer for the comment. We have significantly updated the discussion about signal asymmetry, which is partially related to the probe sensitivity but, as the Reviewer rightly suggests, is also associated with the density of states. We performed additional electron-nuclear dynamics within the Ehrenfest ansatz as implemented in the DFTB+ code to substantiate our claims and provide a qualitative understanding of what is at play (see new figure S10). Additionally, we performed data analysis of carrier population and energy for the hot electrons (new figure S11), similar to what was done for the holes (figure 4C). The hot electron signal is lower due to probe sensitivity; consequently, the error is more significant. However, electron population and energy have a similar shape over time, which is distinctive of what was observed with hot holes. This corroborates the influence of the DOS in the observed signal asymmetry.

Action taken: we performed additional time-resolved calculations to provide a qualitative understanding of what happens to electrons and holes (see new figure S10). We also added to SI the analysis of hot electrons' transient change in population and energy (see new figure S11). We updated the manuscript text with the statement below and several references for support:

The signal asymmetry is related to the L_3 -edge transition XANES's higher sensitivity to the formation of empty states in the d-shell, i.e., the hot holes. However, the asymmetry of excited electrons and holes is also anticipated because of the difference in electron and hole density of states, which is consistent with experimental and theoretical reports. Temporal analysis of the relative changes in mean energy distribution and population for hot holes (Figure 4C) and electrons (Figure S11) reveal a contrasting behaviour. Despite the significant experimental errors related to the lower sensitivity of the XANES probe to occupied states, the hot electrons energy

distribution and population have similar dynamics, peaking in intensity around 100 fs (Figure S11). Electron-nuclear dynamics calculations within the Ehrenfest ansatz, as implemented in the DFTB+ code, were performed to rationalise the observed asymmetry. Figure S10 shows a clear asymmetric electron and hole dynamics, owing to the asymmetric DOS around the Fermi level with the range of the laser energy. These observations are in qualitative agreement with the experimental measurements. The higher localization of electrons in respect to the holes favours carrier multiplication impact excitation that increases the number of carriers and simultaneously reduces their energy.

(Related to comment 3 and figure 3A) While there is an attempt to correlate density of states with experimental data, there is no mention of it in the main text or in figure caption. Since, the experiment and related discussion is focussed around resonant transitions, some text related to density of states could provide more insights.

Reply: We thank the Reviewer for the comment. The comment was largely answered in the previous reply.

Typically, pump fluence plays an important role in the pump-probe experiments. However, in the current data the role of fluence is not evident. Is all data taken at the same pump fluence? If yes, the authors should state this. Additionally, it would be better if authors could comment on how their conclusions would change at lower or higher pump fluence.

Reply: We thank the Reviewer for the comment. The Reviewer is correct that laser fluency is essential for the experiments. We kept the same fluence for all the time-resolved XANES experiments, namely 98 mJ/cm². The value was decided as a compromise between sufficient signal level and absence of saturation in the TR-XAS data (new figure S8). More importantly, the value is in the ~ 2000 K temperature plateau reached around 5-7 mJ/cm² laser pump excitation (see new figure 2C). We also showed that probe fluence did not induce additional effects (see new figure S7).

Action taken: we added two new figures to SI (fig. S7 and S8) and two to the manuscript (Figures 2 B and 2C). We have also updated the manuscript text with the statements:

To exclude additional effects during the time-resolved XAS experiment, we carried out test measurements at two different X-ray fluxes, with a two-fold increase in the flux. It should be noted that the average X-ray flux at the sample position at 11900 eV (monochromatic beam) was about 5×10^9 photons/pulse, corresponding to 2.5×10^{13} W/cm² at applied experimental conditions, i.e. 75 fs X-ray pulse length and $60 \times 60 \mu\text{m}^2$ spot size. The transient XAS spectra measured at 100 fs time delay and two different X-ray fluxes equal to c.a. 3.7×10^9 and 7.5×10^9 photons/pulse, respectively, are plotted in Figure S7. As can be seen, no detectable differences are observed between the spectra, indicating the absence of any nonlinear or multiphoton X-ray interactions. In addition, it should be emphasised that to induce any nonlinear interaction in the hard X-ray regime, fluences in the 10^{18} - 10^{20} W/cm² range are required, as reported in the literature data. Therefore, the experimental tests align with the literature data and indicate that applied experimental conditions were orders of magnitude lower than those required for multiphoton ionisation or nonlinear processes.

Another important aspect is the pump laser fluency used for the TR-XAS experiments. Figure S8 shows the fluence dependence of the TR-XAS signal at an incident energy of 11916 eV and a pump-probe delay of 100 fs. The signal overlapped with the fitted electron temperature estimated from TAS τ_{e-ph} and equation 2 (Fig. 2C). It is clear that the TR-XAS signals are all within the ~ 2000 K temperature plateau was reached around 5-7 mJ/cm² laser pump excitation. Therefore, as a compromise between sufficient signal level and absence of saturation, the TR-XAS measurements were carried out with a laser fluence of 98 mJ/cm².

The temperature for TR-XAS measurements should be mentioned in the text and in the supplementary information.

Reply: We thank the Reviewer for the comment. The time-resolved XANES experiments were performed using a liquid jet, enabling shot-to-shot refreshment of the sample, eliminating the possibility of heat buildup due to charge recombination. Moreover, the experiments were performed in an environmentally controlled room with stability better than ± 0.5 °C, ensuring that the sample solution is isothermal at around 18 °C.

Action taken: we updated the manuscript text with the statement:

To prevent the excitation of damaged Au NPs induced by intense XFEL pulses, a liquid jet was employed to circulate the Au NPs suspended in water. The colloidal solution was refreshed every four hours. The experiments were performed in a climate-controlled laboratory, which, in conjunction with sample circulation (i.e., reduction of local heat deposition), ensured that experiments were performed under isothermal conditions.

The authors use big enough nanoparticles such that electronic structure is the same as bulk. What would happen if smaller nanoparticles are considered? How important are finite size effects? It would be helpful if authors could address this at least qualitatively.

Reply: We thank the Reviewer for the comment. We performed additional electron-nuclear dynamics within the Ehrenfest ansatz as implemented in the DFTB+ code to show that finite-size effects can contribute to the signal if the particles are less than 2 nm (quantum limit of the plasmon). It is clear from the new figure S9 that under such limiting sizes, the electrons and holes are localised primarily within 1 eV from the Fermi level when excited with 2 eV photons. The dynamics are also very similar for both types of charge. We opted for a particle size (ca. 7 nm) well above the finite-size limiting factors. Moreover, the particle size offers a good compromise between sample stability under the XFEL beam and having dipole resonance. Therefore, we expect the findings to apply to a particle with sizes above the plasmon quantum limit and below the multipole resonance.

Action taken: we performed additional electron-nuclear dynamics within the Ehrenfest ansatz as implemented in the DFTB+ code to provide a qualitative understanding of what happens when we use particles where finite-size effects can occur (see new figure S10). We updated the manuscript text with the statement below and several references for support:

The temporal evolution of the hot carrier population and their energy distribution after optical excitation data can be rationalised, excluding the presence of finite-size effects. As expected from the DOS plots in Fig. S9, the electron and hole band occupations will behave differently with changing sizes. The smaller NP (Au₂₅) excitation leads to more well-defined peaks than the larger sizes (Figure S10). This shows that finite-size effects can play a significant role but only when particles are below the plasmon quantum limit (< 2 nm). Opting to use particles with a 7 nm average diameter avoids the presence of finite-size effects, ensures dipole resonances dominate, and guarantees good sample stability to the XFEL beam. This means the findings apply to particle sizes above the plasmon quantum limit and below the multipole resonance.

Reviewer #2:

This is a very interesting manuscript describing advanced time-resolved spectroscopy of hot carrier distributions produced by optical excitation of plasmons. The experiments involve an ultrafast optical pump (75 fs) and a pulsed (<50fs) X-ray probe using the output of a hard X-ray free electron laser facility (specifically the Swiss XFEL). These are very challenging experiments that were carried out very well. The resulting time-resolved spectra show both the hot hole and hot electron energy distributions as a function of time. This is a timely topic as many proposed applications of plasmonic energetic carriers, such as energy conversion with improved efficiency or driving photocatalytic processes, rely upon a clear understanding of the carrier energies produced and the timescale in which they are present. While the experimental approach is excellent, I do have a few observations/suggestions to improve the manuscript.

Reply: We thank the Reviewer for recognising the work's worthiness and the time dedicated to the revision. Also, we thank the Reviewer for the valuable comments, which we addressed below and in the manuscript. The revised version is more precise and has a higher scholar level, thanks to the Reviewer's comments. We hope the provided clarifications and additions removed the Reviewer's initial reservations, and thus, we can get endorsement for publication.

1. I was not able to find the experimental temporal response function? The pump and probe pulse widths are given as well as a detailed description of the fitting to extract temporal decay times, but doesn't this require knowledge of an overall response function? Other experimental factors besides pulse widths sometimes make the temporal response function considerably longer than the individual pulse widths of pump and probe. In an optical pump-probe experiment for example, one would make a cross-correlation measurement between pump and probe, or perhaps perform pump-probe on the pure solvent to try to extract the experimental response function. Since some of the conclusions of the manuscript regarding kinetics of hot carriers, such as Landau damping of ~25fs and maximum hot carrier population at 105fs, it seems important to establish the experimental response function.

Reply: We thank the Reviewer for the comment. This is indeed a very critical aspect. The Landau damping time was estimated from the onset of the rising function used to fit the kinetic traces, i.e., time zero. To increase the accuracy and assess the error for the revised version of the manuscript, we performed fittings at multiple incident photon energies and reported the average value and error. Since we know the changes signal are related to carriers (not shifts in Au Fermi level (Figure 3B), the process's onset is associated with Landau damping. We adopted a similar procedure to estimate the time that we attained the population maximum. The new transient absorption measurements provide further corroboration for values since we fitted an electron-electron scattering time within our experimental resolution (ca. 100 fs). This process happens after Landau damping and before the population maximum is reached. The use of an average over several fittings and the additional TAS provide higher confidence in the reported values.

Action taken: we updated the manuscript text with the statement:

The plasmon Landau damping time can be extracted from the onset of the XAS spectral shape changes caused by hot carrier formation, which started at $\sim 24.6 \pm 10$ fs after the optical excitation. The value is close to what has been previously calculated value. The Landau damping time and associated error were estimated from the average of the onset of the rising function fitting (i.e. time zero) for hot electrons and hot holes.

Following plasmon Landau damping, the hot carriers reach a maximum carrier population at 105 ± 8 fs, estimated from rising edge analysis performed for hot electrons and hot holes. The values are consistent with the optical measurements that revealed that the initial electron-electron scattering convoluted with the rise of the transient absorption signal occurs within ~ 110 fs, corresponding to the average value of the five measured fluencies. The initial electron-electron scattering occurs after the Landau damping quantum mechanical process, with the maximum populations expected to occur shortly after.

2. I believe the following statement of the authors is too strong: “This consensus description of hot carrier formation has been theorized from physical models but is yet to be validated experimentally, partly due to the lack of element-specific techniques with sufficient temporal resolution.” The authors in the next sentence cite two older (but very nice) femtosecond experimental attempts from 1995 and 2000. However, there are probably at least dozens of optical ultrafast pump-probe experiments from much more recent times that have helped us to understand hot carrier dynamics in plasmonic systems, and at least a few more should be cited here. While the authors do have a really novel and advanced time-resolved approach, to me it is a significant misnomer to use the terminology of “yet to be validated experimentally.”

Reply: We are grateful to the Reviewer for their insightful comment, which has helped us to clarify our research further. We fully agree with the Reviewer's observation that optical experiments, by their nature, do not distinguish between carriers and require certain assumptions during the fitting process. This point was effectively highlighted by Heilpern et al in their Nature Communications (2018) 9:1853, where they stated that: *Typically, non-thermal and thermal carrier populations in plasmonic systems are inferred either by making assumptions about the functional form of the initial energy distribution or using indirect sensors like localized plasmon frequency shifts.*

Having stated this, we updated the text to better convey the message we were trying to transmit and included several more references to transient optical measurements. Additionally, we performed our own TAS measurements at different powers to complement our time-resolved XANES and support the need for a sensitive technique to the carriers.

Action taken: we performed additional TAS measurements in figures 1C, 1D, 2B and 2C. We updated the manuscript text with the statement below and several references for support:

The mechanism for hot carrier formation and subsequent thermalisation upon localised surface plasmon resonance (LSPR) excitation is summarised in Figure 1A, completed with hypothesised timescales for each process. Briefly, the electric field of light induces a coherent excitation of Au valence electrons. The decoherence of the plasmon-induced photoabsorption in noble metals can result in non-thermal electron distributions through intraband transitions, aided by phonon (or defect) scattering or diagonal transitions caused by Landau damping or surface collisions, a process expected to take 10-100 fs. Initially, photon absorption triggers an electron energy distribution departing from equilibrium, resembling a double-step-like function in changes to electron occupancy. Over time, successive energy redistributions occur as energetic electrons scatter with themselves, eventually leading to a high-temperature Fermi-Dirac distribution within hundreds of femtoseconds. Subsequently, electron-phonon interactions gradually reduce this electronic temperature over a picosecond time frame. This consensus description of hot carrier formation has been theorised from physical models and supported a variety of pump-probe optical spectroscopy. Usually, an infrared pump laser pulse energises carriers in the conduction band. In contrast, a shorter-wavelength probe pulse tracks the time-dependent changes in differential transmission (or reflectivity), which are associated with the generation and relaxation of excited electrons at interband transition energies.

3. Along the same lines as in point 2, optical pump-probe experiments have already successfully shown the ability to extract hot hole and hot electron energy distributions in plasmonic systems with similar features as shown in the current manuscript. See for example: T. Heilpern et al., “Determination of hot carrier energy distributions from inversion of ultrafast pump-probe reflectivity measurements,” Nat. Comm. 9, 1853 (2018). In this paper, Figure 5b for example shows the hole and electron carrier distributions extracted from the pump-probe data. Thus, while the current experimental approach is very useful and powerful (and element specific), I have concerns that readers may not realize that other ultrafast pump-probe experiments have already shown the ability to extract hot hole and electron carrier distributions.

Reply: We agree with the Reviewer's comment as stated in point 2. The suggested paper is one of the first to show electron and hole distribution from the inversion of ultrafast pump-probe reflectivity measurements. For starters, their model was applied to gold films because acquiring

good reflectivity data for colloidal samples is a significant challenge. Their model builds on the two-temperature model and produces suitable fittings of their data. Still, the model assumes that carriers multiply solely via electron impact, leading to Fermi-Dirac distribution at elevated temperatures within 500-1000 fs (consistent with our measurements). This is why the observed increase in the hot hole population with the energy around -0.35 eV was interpreted as noise, not an indication of the Auger heating mechanism. Our method does not require a model to fit the data since it directly detects carrier concentration and energy. Nevertheless, we appreciate the Reviewer for bringing this paper to our attention, which we used to strengthen our findings further.

4. The data shown in Figure 3 are very interesting for a couple of reasons pointed out by the authors. First, that carrier energies are created far beyond the pump photon energy, and second that the sample does not reach a Fermi-Dirac energy distribution even after 500fs. These are both very significant, impactful conclusions. I believe the authors are saying the fact that carriers with different energies (Figure 3D) have similar amplitude dependence with time shows a non-Fermi-Dirac behavior. However, my feeling is that in order to prove this, the authors should plot the time-resolved spectra at longer times. Sooner or later the change in amplitudes of the different energies within the distribution will begin to show Fermi-Dirac behavior? Since it is a very, very big result to say that non Fermi-Dirac behavior continues to 500fs (typical beliefs are that this is less than 100fs), it would seem that the authors need to show that this experimental approach can observe a Fermi-Dirac behavior at later times. The authors report thermalization at 1.5ps, so some spectra between 500fs and 1.5 ps would really help support the conclusions of the manuscript.

Reply: We are grateful to the Reviewer for their insightful comment, which helped us clarify our research further. We start with the fact that non-Fermi-Dirac behaviour continues to 500fs. It has been suggested that on extended metal surfaces such as Au films, this might take up to 1 ps (Fann et al. Phys. Rev. B 46, 13592 (1992), Heilpern et al. Nature Commun. 9,1853 (2018)). On Au NPs, the Reviewer is correct in stating that this is believed to happen within 100 fs (Inouye et al. Phys. Rev. B 57, 11334–11340 (1998), Baffou & Rigneault, Phys. Rev. B 84, 035415 (2011)). However, analysis of carrier decay with different energies (Figure 4D (former 3D) and new Figure S11) shows that they decay at relatively similar rates, suggesting that the high-temperature Fermi-Dirac takes longer than 100 fs. The precise time determination requires time-resolved high-resolution XAS and XES, which were unavailable. However, as the first indication, our data supports forming such a state after 500 fs because we start detecting a narrowing of the energy distribution after 250 fs.

The complete electronic thermalisation was established by fitting the signal decay for hot electrons and holes, which coincided with the time at which the TR-XAS disappeared. XANES is an atomic spectroscopy sensitive to electron populations. Consequently the disappearance of the transient signal is indicative of the absence of hot carriers. This contrasts with the TAS data where the signal of hot carriers is convoluted with phonons, which means one extracts the components through a data fitting procedure. However, the determined time for complete thermalization is consistent with the TAS τ_{e-ph} .

Action taken: we updated the manuscript text with the statements below and several references for support. We also added a new figure to SI (figure S11)

The final aspect related to the plasmon carrier relaxation is establishing the time when the Fermi-Dirac distribution is reached. For extended metal surfaces (e.g. Au thin films), time-resolved studies suggest forming a Fermi-Dirac-like distribution characterised by a sizeable effective electron temperature within 1 ps. However, in Au nanoparticles, the electronic gas is expected to thermalise very fast to a Fermi-Dirac distribution over a time scale $\tau_e \sim 100$ fs. In such a scenario, one is to expect carriers with different energies to decay at different rates, with the ones with the highest energies decaying more rapidly. Figure 4D (hot holes) and S12 (hot electrons) show carriers' decay with different energies over a 500 fs. The hot carriers decay at a similar rate independent of their energy, suggesting that forming a Fermi-Dirac-like distribution

characterised by a sizeable effective electron temperature takes longer than 100 fs, as proposed. To establish the exact time, one must perform time-resolved high-resolution XAS and XES, which were not accessible at the time of this measurement.

On the complete thermalisation:

The complete electronic thermalisation occurring within ~1.5 ps, consistent with the τ_{e-ph} of about 4-5 ps established with TAS and confirmed the ultrafast hot carrier relaxation as the primary bottleneck limiting plasmonic applications.

Reviewer #3:

Wach et al describe a fs-resolved XAS study on plasmon-excited Au nanoparticles in solution. They find an increased absorption below the Au L3 X-ray absorption edge, which they assign to the induced absorption of “hot” holes. They fit this absorption feature with a simple Gaussian model and compare it to the valence-XPS spectrum, concluding that the energy distribution is much broader than the photon energy employed in the plasmon excitation process. This leads them to conclude that an Auger heating process dominates the relaxation of hot carriers. While the overall story sounds quite impressive, the conclusions drawn are not supported by the data. The data analysis is rather simple and control experiments are not presented. While I believe that the authors are well-aware of the intricacies of these kind of measurements (large laser fluences, nonlinear excitation, broadening, etc.), they do not provide sufficient data to justify their approximations and assumptions. I am also missing a careful comparison of the findings to what is known from the large body of literature on ultrafast carrier dynamics in Au nanoparticles. Overall, I do not recommend publication of this manuscript in its present form.

Reply: We want to thank the Reviewer for recognising the work's worthiness and the time dedicated to the revision. Also, we thank the Reviewer for the valuable comments, which we addressed below and in the manuscript. Thanks to the Reviewer's comments, the revised version is more precise and has a higher scholar level. More specifically, we added additional data concerning laser pump fluencies and the possible role of X-ray probes in inducing non-linear processes. We also carried out further data analysis. Furthermore, we perform time-resolved absorption spectroscopy (TAS) to support the TR-XAS data and time-resolved theoretical calculations to evaluate possible finite-size effects and to help explain the observed asymmetry in the hot hole and hot electron signal. We hope the provided clarifications and additions removed the Reviewer's initial reservations, and thus, we can get endorsement for publication.

- The authors are not clear about the various processes occurring upon plasmon excitation and their respective time scales. In particular, they entirely avoid the mention of electron/hole-phonon coupling, which is known to be the primary relaxation channel for carriers in metallic nanostructures taking place on the 0.5-few ps time scale. The authors, however, seem to associate the decay of their XTA signals to “thermalization” of the carriers, not carrier-phonon coupling. The latter results in a population of phonons and lattice thermalization through phonon-phonon scattering on the tens of ps time scale. These processes should also be visible in the XAS spectrum, as a previous ps-resolved XAS study on Au nanoparticles has shown (Zamponi, F. et al. Probing the dynamics of plasmon-excited hexanethiol-capped gold nanoparticles by picosecond X-ray absorption spectroscopy. *Physical Chemistry Chemical Physics* 16, 23157–23163 (2014)). Why is this not visible in the XFEL data? How do the XFEL and synchrotron data compare? They were measured with similar excitation conditions.

Reply: We thank the Reviewer for the comment. This XFEL study covered the time domain when hot carriers are present, i.e., before their thermalisation. The XANES part of the XAS measurements cannot access the coupling between electrons and phonons. To gain information about this, we need to perform EXAFS measurements as Zamponi et al. did. Zamponi et al. showed that on a 100 ps time scale, the released heat from lattice thermalisation induces a structural change in the Au NPs, which is well-known and understood. This is also consistent with the phonon-phonon lifetime (τ_{ph-ph}) measured by our TAS experiment. The current gap knowledge is in what occurs before the coupling: carrier formation dynamics, energy distribution and multiplication mechanism.

The steady-state XAS spectra of the Au NPs measured at the XFEL and synchrotron are identical in shape and energy (see Figure S4).

Action taken: we updated the manuscript text and added TAS data supporting the time-resolved XAS data presented.

- The authors should carefully and rigorously place their results in the context of the enormous body of work on ultrafast spectroscopy of plasmonic nanostructures. They should address in detail how their excitation conditions (~ 100 mJ/cm² incident fluence) influences the observed dynamics and is relevant for plasmonic chemistry as described in the motivation of the work. The authors state that by choice of wavelength the excitation of interband transitions is avoided. This may be true in a low-excitation/perturbation regime. However, the used laser fluence is so large that two- or multi-photon absorption cannot be avoided. The authors should therefore present a careful laser fluence dependence quantifying the amount of multi-photon absorption (or indeed showing that it is negligible, as they write). In the SI they write: “The pump laser fluence (98 mJ/cm²) was determined at the beginning of the experiment and chosen to maximise the excited-state fraction while minimising multiphoton absorption effects.” And in the main text: “Note that the low optical laser fluency and short pulse duration used in this experiment make it unlikely that multiphoton excitation of single electrons occurs.” But no data supporting this statement was provided! In fact, a fluence of 100 mJ/cm² is really high if compared to typical fluences used for transient optical absorption experiments in the literature (< 1 mJ/cm²). And the laser excitation pulses (75 fs) are quite short - the shorter the laser pulse, the larger the probability of multi-photon absorption. Importantly, it is known from the literature that the decay dynamics of hot carriers in plasmonic nanomaterials depends on the excitation density (e.g. J. Phys. Chem. C 2023, 127, 43, 21176–21185). A laser fluence dependence study is thus crucial to place the results in the context of existing work in the literature. An ultrafast optical transient absorption data set taken under identical excitation conditions would also be helpful to compare the results presented to those in the literature.

Reply: We thank the Reviewer for the comment. The Reviewer is correct in stating that laser fluency affects the plasmonic carrier dynamics, as shown in, e.g. J. Phys. Chem. C 2023, 127, 43, 21176–21185). To establish that X-ray experiments are valid and consistent with published optical data, we performed power dependence TAS measurements (new Figures 1C, 1D), which enables us to extract the electron-phonon lifetime (τ_{e-ph}) as a function of laser fluency (new Figure 2B). This permitted us to connect the optical with the X-ray measurements (new Figure 2C). The electronic temperature reaches a temperature plateau of ~ 2000 K when exciting the sample above 5-7 mJ/cm² (see new figure 2C). We used 98 mJ/cm² excitation to compromise sufficient signal level and absence of saturation in the TR-XAS data (new figure S8). We also showed that probe fluence did not induce additional effects (see new figure S7). Collective excitation of the electron gas requires multiphoton absorption, which is significantly helped by plasmonic large cross-sections. Nearly all published transient data was performed under such conditions. However, what we think the Reviewer highlights relates to the multiphoton excitation of single electrons. Based on our measurements, this is not taking place because this would result in higher electronic temperature. According to Atwater and coworkers’ deviations from a temperature-independent electron-phonon coupling constant or nonlinear temperature dependence of the electron heat capacity were predicted for electron temperatures significantly exceeding 3000 K for gold (Phys. Rev. B 94, 075120 (2016); Phys. Rev. Lett. 118, 087401(2017)), which is not the present case.

Action taken: we updated the manuscript text and added power-dependent TAS data supporting the time-resolved XAS data presented.

- One of the striking findings of this work is the fact that the width of the induced absorption feature below the edge spans ~ 5 eV, i.e. much more than the employed photon energy of 2.3 eV. The authors write: “However, this does not explain the unique observation of carriers with energies above the photon energy, even considering the energy broadening induced by 5.41 eV Au the L3-edge core-hole broadening, which limits the experimental energy resolution.” And “Note the energy of the holes is affected by the Au core-hole lifetime.” No data/analysis is presented to backup these statements. How DID the authors consider the core hole life time broadening? Clearly, this will have a big affect in the transient spectrum and without considering it, how can a realistic energy distribution of hot carriers be extracted at all? Similarly, the authors

make an attempt to quantify the carrier multiplication by comparison of the transient XAS spectrum magnitude at the Au L3 edge with the white line feature in the static Pt L3 edge spectrum. Without taking into account the differences in core-hole life time between these two edges, this analysis seems faulty. Finally, the authors extract changes in the energy distribution by fitting the transient spectra to a superposition of two Gaussians. They focus on the changes in the Gaussian peak fitting the positive spectral feature (that they assign to absorption into hole states), but do not consider what happens to the negative feature (supposedly representing hot electrons). The dynamics of both are intertwined due to the large core-hole life time broadening at the Au L edge. It thus seems wrong to just focus the analysis on one side of the spectrum, not discussing the other side...

Reply: We thank the Reviewer for the comment. According to Krause et al. (J. Phys. Chem. Ref. Data 8, 329-338 (1979)) the core-hole lifetime of Au L3-edge is 5.41 eV while the Pt is 5.31 eV. This slight difference is insufficient to induce significant changes in spectral shape and, thus, affect our estimations meaningfully. Therefore, the approach used to estimate the carrier involvement in qualitative terms is valid. The qualitative result supported non-radiative decay as a significant channel, which would remain so even if we were off by a factor of two. That is certainly not the case.

The second aspect is the effect of core-hole lifetime in broadening the signal. Our statement emphasises that the exact carrier energy cannot be deduced from the data because the core-hole lifetime broadens the energy scale. However, the induced broadening is the same for all the hole energies. Thus, one can use the changes in energy distribution to deduce what carrier multiplication processes are at play. In the case of carrier multiplication via impact excitation (the most common method), one should expect the carrier energy distribution to decrease after the first generation of carriers is formed (around 100 fs after excitation), independent of the induced broadening. The fact that this wasn't observed indicates that another possible mechanism is also participating, namely, Auger heating. Precisely determining carrier energy requires time-resolved high-resolution XAS and XES, which were unavailable.

Action taken: we updated the manuscript text with the statements below and several references for support. We also added a new figure to SI (figure S12)

The final aspect related to the plasmon carrier relaxation is establishing the time when the Fermi-Dirac distribution is reached. For extended metal surfaces (e.g. Au thin films), time-resolved studies suggest forming a Fermi-Dirac-like distribution characterised by a sizeable effective electron temperature within 1 ps. However, in Au nanoparticles, the electronic gas is expected to thermalise very fast to a Fermi-Dirac distribution over a time scale $\tau_e \sim 100$ fs. In such a scenario, one is to expect carriers with different energies to decay at different rates, with the ones with the highest energies decaying more rapidly. Figure 4D (hot holes) and S12 (hot electrons) show carriers' decay with different energies over a 500 fs. The hot carriers decay at a similar rate independent of their energy, suggesting that forming a Fermi-Dirac-like distribution characterised by a sizeable effective electron temperature takes longer than 100 fs, as proposed. To establish the exact time, one must perform time-resolved high-resolution XAS and XES, which were not accessible at the time of this measurement.

Second round of revision:

Reviewer #2:

I believe the authors have worked very hard to address the reviewer comments. They included additional computational efforts (particularly in the supplemental information) and performed additional significant experiments, including performing their own transient absorption spectroscopy (Figures 1c, 1d) in support of their ultrafast X-ray absorption spectroscopy results. They also significantly improved placing their experiments/conclusions within the context of past transient absorption spectroscopy efforts on hot carrier dynamics. They have also added clarification for their interpretation of the Landau damping mechanisms by more carefully addressing all possible mechanisms. Overall, I believe the manuscript is suitable for publication.

Reply: Thank you for your thorough and thoughtful assessment of the manuscript. We appreciate your recognition of the extensive efforts made by the authors in response to the Reviewers comments. The additional computational work, particularly in the supplemental information, alongside the new experimental data, such as the transient absorption spectroscopy in Figures 1c and 1d, indeed represent a significant contribution to supporting the ultrafast X-ray absorption spectroscopy results. The enhanced contextualization of our findings within the broader scope of transient absorption spectroscopy on hot carrier dynamics, as well as the detailed clarification of the Landau damping mechanisms, demonstrates a comprehensive and rigorous approach to addressing the concerns raised. Your endorsement underscores the substantial improvements made, and we concur that the manuscript is now well-suited for publication.

Reviewer #3:

I have looked at the replies and the revised manuscript. While I see some improvements, I still do not think this should be published in Nature Nanotechnology.

Reply: We strongly disagree with this assessment, which also contradicts what Reviewer 2 stated, namely “They included additional computational efforts (particularly in the supplemental information) and performed additional significant experiments, including performing their own transient absorption spectroscopy (Figures 1c, 1d) in support of their ultrafast X-ray absorption spectroscopy results. They also significantly improved placing their experiments/conclusions within the context of past transient absorption spectroscopy efforts on hot carrier dynamics. They have also clarified their interpretation of the Landau damping mechanisms by more carefully addressing all possible mechanisms.” The manuscript and supporting information regarding content and data were extensively updated.

The study is not complete - the conclusions drawn (non-Fermi-Dirac distribution beyond 500 fs) are not sufficiently supported by the data.

Reply: The reviewer's comment represents a mischaracterization of our measurements and findings. The formation of a non-Fermi-Dirac distribution of carriers is a well-established process, expected to occur within approximately 100 fs, aligning with our observations of carrier dynamics measured up to 1.5 ps for several energy levels above and below the Fermi level, as illustrated in Figures 3C and 3D. The reviewer appears to conflate the formation of this non-Fermi-Dirac carrier distribution with carrier multiplication processes, which commence immediately after the initial hot carrier population is established, approximately 25 fs post-excitation. Our analysis of the energy distribution of the signal at various time delays was aimed at elucidating the carrier multiplication mechanism. Given that carrier multiplication processes are generally believed to cease after two to three generations, within several hundred femtoseconds (as detailed by Khurgin in *Nanophotonics* 9, 453-471 (2020)), we focused our analysis on the initial 500 fs, a decision informed primarily by signal-to-noise considerations. While the prevailing literature suggests that carrier multiplication occurs solely via impact excitation, our data show a continued broadening of the carrier energy distribution long after the maximum carrier population is reached - specifically around 150 fs, following the peak population at 100 fs. This suggests the involvement of the Auger excitation process, a phenomenon that, to our knowledge, has not been previously reported.

The authors mention additional experiments (high-resolution XAS and XES) that are needed to support the interpretation. I agree with that. I understand that XFEL experiments are precious and one cannot just go back to the lab and remeasure, but I don't think this should lower our bar of excellence to publish in a prestigious journal like Nature Nanotechnology.

Reply: The manuscript sought to elucidate hot carrier dynamics and the mechanisms underlying their multiplication using a spectroscopic probe that is selective to the carriers and, therefore, unaffected by confounding factors such as thermal effects and near-field interactions, which often complicate the interpretation of ultrafast optical data (a commonly used approach). Determining the precise energy of the generated carriers was beyond the scope of this study. Achieving this would require high-resolution X-ray absorption spectroscopy (XAS) and X-ray emission spectroscopy (XES), specifically high-energy off-resonant XAS (a technique pioneered by some of the authors) and valence-to-core XES. However, both methods typically suffer from extremely low signal-to-noise ratios, necessitating either high probe flux or high repetition rate measurements (in the MHz range). Given that the XFEL flux we employed was intentionally kept below the threshold where nonlinear X-ray processes, such as two-photon or saturable absorption, might occur, our only option is to perform high repetition rate measurements, which are currently being developed at the European XFEL but are not yet operational. Nonetheless, such measurements would not contribute additional insight into the dynamics and multiplication mechanisms reported in the manuscript, which are more accurately captured using direct XANES absorption. This

technique provides the highest measurable signal, making it less susceptible to artifacts, such as XFEL beam instability, thus offering a more reliable characterization of the carrier dynamics.

My criticism on the comparison with ps-resolved synchrotron data and the fluence dependence are not addressed adequately/fully.

Reply: We strongly disagree with this statement due to its flawed premise. Specifically regarding synchrotron experiments, the Reviewer appears to suggest that achieving the low-picosecond resolutions necessary to detect signals from hot carriers is possible. This assertion requires correction. Pump-probe experiments at synchrotrons typically offer temporal resolutions on the order of 80 ps (as demonstrated by Zamponi et al., *Phys. Chem. Chem. Phys.*, 2014, 16, 23157, cited by Reviewer 2), which is significantly longer than the lifetime of hot carriers (<2 ps). I speak with authority on this matter, having collaborated closely with Prof. Majed Chergui, a pioneer in such measurements, and having been a speaker on this topic at the esteemed RAC International Summer School - a well-regarded forum for advanced materials research at large-scale X-ray and neutron facilities (<https://www.rac-school.org>). Additionally, we have provided supporting information on the fluence dependence of the data measured at the XFEL (Figure S8), complemented by fluence-dependent optical data (Figure 2C), to establish a correlation between the XFEL data and the current understanding of plasmon-induced hot carrier dynamics derived from optical experiments. We are unclear on what further power-dependence data the Reviewer believes is missing, as we think the presented data adequately addresses this aspect.

The authors write that the XANES region is only sensitive to carrier populations, this is not true. It is well-established that the XANES region is also sensitive to the local structure (and thermal/geometric disorder) and as such can be, and has been, used to probe electron-phonon coupling. In fact, there is a large body of ultrafast XANES work on solids (metal oxides, metals) showing this. I can therefore not endorse publication of this manuscript for this journal.

Reply: We believe that our statement is being taken out of context. According to textbook definitions, XANES is primarily sensitive to electronic changes, while the extended region (EXAFS) is sensitive to local structural changes. Major structural modifications, such as reductions or oxidations, can significantly influence the XANES spectrum. However, more subtle geometric distortions, such as those induced by thermal effects, are considerably more challenging to detect—this is the point we aimed to convey. Our data support this interpretation, as the transient XANES signal we measured vanishes within 1.5 ps, coinciding with the disappearance of the hot carriers. In contrast, the thermal component, with a lifetime of approximately 50 ps, would be expected to persist in the data if XANES were highly sensitive to it. We are not claiming that detecting these effects is entirely impossible, but achieving this would require a significantly higher signal-to-noise ratio or the use of EXAFS techniques (e.g., as demonstrated by Zamponi et al., *Phys. Chem. Chem. Phys.*, 2014, 16, 23157, cited by Reviewer 2), which is incompatible with the conventional XFEL operation mode currently in use.

Answers to the Reviewers comments:

Reviewer #1:

This article presents an ultrafast X-ray absorption study of colloidal Au nanoparticles excited via their plasmon resonance. The use of X-ray absorption is ideal here as it allows a detailed visualization of the electronic structure changes incurred by the system after photoexcitation. The main conclusions are a commensurate hole and electron relaxation, a finite risetime for the population of charge carriers, which the authors interestingly attribute to the occurrence of Auger processes. The paper deserves publication but it contains a number of statements, which the authors ought to modify.

Reply: Thank you for your time and thoughtful evaluation of our manuscript. We greatly appreciate your positive feedback regarding the study's significance and the potential of X-ray absorption to elucidate the electronic structure changes in photoexcited colloidal Au nanoparticles. We are pleased that our findings on electron and hole relaxation dynamics and the role of Auger processes resonated with you. In response to your constructive comments, we have carefully reviewed and revised the manuscript to address all points raised. We believe these changes have strengthened the clarity and depth of our analysis, and we hope that the revisions meet your expectations. We sincerely thank you for your support and strong endorsement for publication. Please find our detailed responses to each of your comments in the revised manuscript.

- bottom of p. 2 and top of p. 3, the authors introduce XAS, but the last 2 sentences of this § speak about optical excitation, they then jump back to XAS in the following §.

Reply: We thank the Reviewer for the comment. Indeed the mentioned sentence (“Hot carriers emerge from the interaction between external electric fields and valence electrons, creating electrons and holes with energies above and below the Fermi level (E_F).”) was misplaced. In the revised manuscript it was placed in a different place, where it fits the context better.

Action taken: we have moved the sentence to a different part of the manuscript (page 3).

- p. 3, 2nd §: the authors mention a time resolution of 5 ps for synchrotrons. While this is achievable, it is rather uncommon. The temporal resolution is more around 50-100 ps.

Reply: We appreciate the Reviewer's remark and fully agree that the typical temporal resolution is indeed in the range of 50-100 ps. However, in a previous revision, we were advised to include details on the mode tested at the Argonne synchrotron, which can achieve resolutions close to 5 ps. We opted to mention this mode as a reference, acknowledging that even with this high resolution, capturing the initial stages of hot carrier formation, multiplication, and relaxation through phonon coupling would remain challenging (to not say impossible), as discussed in our manuscript.

Action taken: The temporal resolution has been revised to reflect the Reviewer's suggestion, aligning with standard synchrotron operation for improved consistency.

- p. 7, 1st §: the authors conclude their discussion on the Temperatures, started the page before, but they do not conclude so a logical link can be made with the following §.

Reply: We thank the Reviewer for this valuable comment. We agree that the original discussion lacked a clear connection to the subsequent text. To address this, we have added a concise linking sentence to enhance the flow and provide seamless connectivity to the following section.

Action taken: we have added the following sentence to the manuscript manuscript:

This temperature analysis demonstrates that the pump fluence used at the XFEL falls within this stagnation regime, leading to the generation of a large number of hot carriers rather than an

increase in carrier energy. This insight is critical for understanding the population of hot carriers under these experimental conditions and provides a basis for analyzing the subsequent electron relaxation dynamics.

- p. 7, 3rd §: The statement "Au NPs have yet to be measured at XFELs" is to the reviewer quite shocking since Au NPs have been among the most studied systems at XFELs. Ultrafast X-ray diffraction, coherent diffractive imaging, X-ray photon correlation spectroscopy, X-ray holography, etc. have been performed at XFELs using Au NPs, which serve almost as the ideal test model.

Reply: We thank the Reviewer for bringing this to our attention. We agree that our original statement lacked precision, and we understand the resulting confusion. Our intention was to convey that, to our knowledge, ultrafast XAS measurements on Au nanoparticles have not yet been performed. This is partly due to the fact that only recently have XFEL facilities started providing the necessary beam energies to excite the L₃-edge of 5d elements. We have revised the statement to clearly convey our intended meaning while ensuring it appropriately acknowledges previous work in the field.

Action taken: we have corrected the statement in the manuscript with the sentence:

Au NPs are commonly utilized as ideal test models in XFEL studies on diffraction and imaging. However, to our knowledge, ultrafast XAS studies on Au nanoparticles have yet to be conducted at these facilities, partly because access to the hard X-ray energies necessary to probe the Au L₃-edge transition has only recently become available.

- Same §, what is a "slight absorption"? You mean a weak absorption?

Reply: We thank the Reviewer for bringing this to our attention.

Action taken: The aforementioned expression has been replaced by a more appropriate one: "weak absorption".

- p. 9, 2nd §: The analogy between the bulk and NP electronic structure for sizes above 4 nm has long been established (see Doremus, J. Chem. Phys. 1964, 40, 2389–2396; Kawabata et al., J. Phys. Soc. Jpn. 1966, 21, 1765–1772). It is sufficient to say that the present results fit into this trend.

Reply: We thank the Reviewer for this comment. We made the statement more direct.

Action taken: we have updated the text with:

The observation that the electronic structure of the Au NPs used in this study matches that of bulk Au was further supported by theoretical calculations of the Au DOS as a function of particle size. These calculations indicate that for particles above 3 nm, the electronic structure of Au NPs begins to resemble that of bulk gold (Fig. S6). Consequently, the similarity in the XAS spectral shapes observed for the Au NPs and bulk Au foil (Fig. S4) is consistent with these findings.

- p. 10: I would move §§ 2 and 3 to the SI. These are checks of the fluence dependence that are not directly relevant to the conclusions of the paper.

Reply: We thank the Reviewer for this insightful comment. While we agree with the Reviewer's suggestion, we would also like to note that this placement was requested in a previous revision. We defer to the editor's discretion regarding the optimal placement of this text. One reason for including it in the main text was to facilitate photon density estimation, which is essential for calculating process efficiency in subsequent analysis.

- Fig. 3: I would overlap the traces in panels C and D after flipping the latter. This will better visualize the correspondance between the 2 traces.

Reply: We thank the Reviewer for this insightful comment. In response, we have prepared the suggested figure and have incorporated it into the manuscript. Additionally, to maintain visual symmetry in the figure, we included a version with signals normalized to their maximum values. This adjustment clarifies the similarity in the dynamics of holes and electrons.

Action taken: we have updated figure 3 as suggested by the Reviewer.

- p. 13: Given the very noisy electron trace (fig. 3D) I am surprised that the uncertainty of the fit is only 65 fs, barely twice that of the hole trace.

Reply: We thank the Reviewer for this comment. We agree that the fitting error for electrons is only about twice that for holes, which might seem surprising given the noise in the electron data. To clarify, due to the noise in the electron data, we constrained the fitting primarily to the decay phase of the process. Specifically, we used the rising parameters derived from the hole kinetic data to aid in fitting the electron trace, as these parameters should be consistent across both datasets. This approach allowed us to achieve a reasonable fit for the decay despite the noise. We have added a brief note to the manuscript to highlight this methodology.

Action taken: we have added to the manuscript test the following note:

Note that, to improve the fit for the hot electron data, which has a significantly lower signal-to-noise ratio compared to the hot hole data, the fitting was focused on the decay portion of the process. The rising parameters, derived from the hole kinetic data, were applied to aid in fitting the electron trace, as these parameters should be consistent across both datasets.

A couple of minor points:

- Be careful at the confusion between kinetics and dynamics. The former relates to population decay, the latter to wave packets. Here the authors deal only with kinetics, but they use "dynamics" in page 13 and fig. 4.

Reply: We thank the Reviewer for bringing this to our attention. We changed the word dynamics to kinetics in page 13 and figure 4.

Reviewer #2:

After carefully reviewing this manuscript, unfortunately I have to conclude that I cannot recommend its publication in its current form. I agree with all the other reviewers that the data are very unique and potentially very interesting. However, in my opinion the study is fundamentally flawed as it does not have any control experiments for the ultrafast measurements. Naturally this leads to many questions that were raised by previous reviewers.

Reply: We thank the Reviewer for the time and consideration in reviewing our manuscript. We appreciate your recognition of the uniqueness and potential interest of our data, as well as your valuable feedback on the need for control experiments in the ultrafast measurements. We understand the importance of including these controls to strengthen the study and address the questions raised by previous reviewers. In response, we revised manuscript that incorporates additional control experiments to validate our findings more robustly. We believe these additions will clarify the results and provide the rigor necessary for a comprehensive understanding of the ultrafast measurements presented. Thank you once again for your thoughtful critique, which has been instrumental in guiding these improvements.

I will highlight only one example here: 532 nm excitation CAN excite interband carriers in gold since bulk gold does absorb in this region. Indeed this has been confirmed in ultrafast measurements as well in many of the papers cited by the authors. The authors might be correct that this is not the main contribution in their experiments, however no logical arguments or back-of-the-envelope estimates can convince me in this unless there is a control experiment on e.g. bulk gold, thin films etc. Without such measurements, the entire argument regarding plasmon dephasing, which is the main premise of the paper, collapses.

Reply: We thank the Reviewer for this insightful comment. As the Reviewer correctly suggests, and in agreement with the published literature (see, e.g., Yamada et al., J. Phys. Chem. C 111, 11246-11251 (2007)), when exciting to the red of the LSPR maximum, the interband contribution to the hot carrier population decreases significantly. This is because the excitation energy becomes insufficient to excite d-electrons into the sp-band (see, e.g., Johnson et al., Phys. Rev. B 6, 4370-4379 (1972); Dreesen et al., Appl. Phys. B 74, 621-625 (2002)). In our study, we used laser pulses with an energy of 2.33 ± 0.13 eV. Considering the Au d-shell location at 2.5–2.58 eV, this implies a low probability of exciting the interband transitions under our experimental conditions. To further substantiate this assumption, we conducted optical TAS measurements at different excitation wavelengths: specifically, interband (below the LSPR peak at 450 nm), resonance maximum (at the LSPR peak at 520 nm), and intraband (above the LSPR peak at 532 nm) (see new Figs. S7 and S11).

A direct comparison of excitation probabilities (Fig. S7) shows that intraband excitation is significantly more efficient at generating hot carriers than interband excitation, which, along with the lower probability of interband excitation, indicates that the interband contribution to our transient XAS signal is minimal. Furthermore, the kinetics of the interband transition differ substantially from those of the intraband (Fig. S11). If the interband transition contributed meaningfully to our transient XAS signal, we would expect a double-exponential decay rather than the observed single-exponential decay. Additionally, the detected thermalization time (~ 1.5 ps) aligns well with the ~ 2 ps reported for predominantly LSPR transitions (see Sun et al., Phys. Rev. B 50, 15337-15348 (1994)) and with the classic two-temperature model (Kaganov et al., Sov. Phys. JETP 4, 173-178 (1957)).

These findings provide strong evidence that interband excitations are largely avoided in our experiments and therefore contribute minimally, if at all, to the transient XAS signals. We acknowledge the Reviewer's recommendation for further control experiments on bulk gold or thin films to strengthen this argument and agree that such measurements would provide additional support. However, given the current data and analyses, we believe that our approach offers a reasonable and robust basis for minimizing the role of interband excitation in our interpretation of the results.

Action taken: we have added to Supporting information figures S7 and S11.

We have added to the manuscript text:

The centre of the Au d-shell is located at 2.5-2.58 eV (~ 496-480 nm) from the metal E_F , meaning that the laser pulse with a 2.33 ± 0.13 eV (15 nm FWHM) photon energy can only excite the low energy tail of the d-shell at best. Therefore, the optical photon energy reduces to a high degree the interband excitations with an absorption onset at 2.38 eV if one ensures that only fundamental dipole LSPR transitions of Au NPs are excited, as controlled by laser fluence. To further confirm that interband excitation does not contribute significantly to the overall signal when exciting to the red of the LSPR peak, optical TAS measurements were conducted at different excitation wavelengths: specifically, interband (below the LSPR peak at 450 nm), resonance maximum (at the LSPR peak maximum of 520 nm), and intraband (above the LSPR peak at 532 nm). Comparison of the kinetic traces extracted near the excitation wavelength (Fig. S7) reveals that pure interband transitions are considerably less efficient than intraband transitions at generating hot carriers. The observed decrease in signal amplitude when exciting beyond the LSPR peak, as opposed to at the LSPR maximum, further supports the reduced contribution of interband excitation for wavelengths selected to the red of the LSPR maximum. Error! Bookmark not defined.

And:

The ability to fit the data with a single exponential decay further supports the argument that interband excitations are largely avoided, as these excitations exhibit distinct kinetic decays (see Figure S11). If interband excitations contributed significantly to the signal, a more complex decay behavior would be expected.

Similarly, other significant claims in the paper are open to alternative interpretations without control experiments. As one of the reviewers mentioned, we all understand that FEL experiments are challenging, and we appreciate the novelty of applying these measurements to Landau damping in plasmonic structures. However, we must uphold basic standards of scientific rigor.

Reply: Thank you for your valuable feedback and for recognizing the novelty of our approach in applying FEL measurements to study Landau damping in plasmonic structures. We understand the critical importance of upholding rigorous scientific standards, particularly within the complex and evolving field of FEL experiments. Regarding the need for control experiments, we appreciate this suggestion and have implemented blanks and validation studies that we deemed feasible within the limited FEL beam time, which, as the Reviewer is likely aware, is highly competitive and scarce. These validation efforts included fluence-dependent studies on both the pump and probe, ensuring that the observed signal is genuine and not an artifact. Additionally, we performed complementary measurements using ultrafast optical spectroscopy and synchrotron techniques to ground the FEL data within the established understanding of the processes, primarily informed by ultrafast optical spectroscopy.

As experienced teams in ultrafast and X-ray spectroscopy, we have designed these control studies to clarify our interpretations and strengthen our conclusions. Based on current methodological constraints, we genuinely believe that the validation and blank experiments we conducted offer a high level of confidence in our findings. Further suggested experiments, such as transient XAS at a synchrotron, are unlikely to provide meaningful data for this study: typical temporal resolutions at such facilities (50-100 ps) are significantly longer than the carrier lifetimes observed (~1.5 ps), making it technically infeasible to capture the signal of interest. Securing transient XAS beam time at a synchrotron is not straightforward because it necessitates a clear scientific question, which remains challenging to frame given the rapid decay of our signal well before the synchrotron can detect it.

We sincerely appreciate your insights and suggestions, which have enabled us to reinforce the rigor of our work to the greatest extent possible under current constraints.

Reviewer #3:

This manuscript describes a high-level experimental effort to apply ultrafast time-resolved X-ray absorption experiments to help determine the dynamics and energy levels of “hot” holes and electrons in plasmonic gold nanoparticles. The topic is well-introduced and the manuscript overall is well-written. I think the work will have strong interest from Nature Communications readers. The authors report hot carrier energies that extend well beyond the more typical limit of the pump photon energy, which will garner significant interest, and may be well justified since this is a new experimental approach to monitor hot carriers.

Reply: We would like to thank the Reviewer for taking the time to carefully evaluate our manuscript and for their thoughtful comments. We are especially grateful for the strong endorsement and recognition of the novelty of our work. We appreciate the Reviewer’s acknowledgement of the significance of our findings, particularly regarding the detection of hot carrier energies that extend beyond the conventional pump photon energy limit. We are pleased that the Reviewer found our manuscript well-introduced and well-written, and we are encouraged that our experimental approach will be of strong interest to the Nature Communications readership. Thank you once again for your valuable feedback and support.

I do have a few small suggestions that if addressed could increase impact:

1. Figure 1a could probably be optimized a bit further. It shows e-ph dynamics extending to 10ns, but I believe the authors may have intended to ph-ph dynamics to the text in figure? It might just be a small omission as 10 nanoseconds is too long for e-ph interactions. The authors say themselves in the text that e-ph interactions occur on a “picosecond time frame.” They also say in the text on page 13 that ph-ph relaxation occurs in “10s of nanoseconds.” I think Figure 1a could just use a little optimization to be more consistent with the text.

Reply: We thank the Reviewer for this helpful comment. In response, we have revised Figure 1a to better align with the known timescales and specific processes discussed in the manuscript. We clarified the labeling to distinguish between electron-phonon (e-ph) interactions, which occur on a picosecond timescale, and phonon-phonon (ph-ph) relaxation, which occurs over tens of nanoseconds, as described in the text. This adjustment should make the figure more consistent with the established dynamics and with the explanations provided in the manuscript. Thank you for drawing our attention to this detail.

Action taken: we have revised figure 1a as below:

2. I am a little confused why the authors say (line 116) that the pump beam in transient absorption is “usually” in the infrared range? Isn’t it usually just chosen to excite plasmon, and for Au nanoparticles that are spherical, that is in the visible? In the work in this paper, the excitation is in the visible at 535nm. I might be missing the point that the authors are trying to make here.

Reply: We appreciate the Reviewer’s insightful observation and apologize for any confusion caused by our wording. We have revised this section in the manuscript to clarify. Our initial statement referred to the classical description of laser pulses of femtosecond laser amplifiers setups, which typically generate pulses in the near-infrared range (e.g., Ti-sapphire at 800 nm or Ytterbium at 1064 nm). These pulses are often modified with an Optical Parametric Amplifier (OPA) to achieve visible excitation, specifically for exciting the plasmon in gold nanoparticles. As the Reviewer correctly notes, our excitation in this work is indeed in the visible range, at 532 nm. We have updated the text to reflect this more accurately. Thank you for helping us improve the clarity of our manuscript.

3. It is best practice in transient absorption experiments to show the kinetics plots extend long enough to show the signal go to zero which helps assure the reader that there is reduced likelihood of artifacts (such as the presence of a long-time signal that is re-excited by the next pulse). If the authors have the longer time data for Figure 3, it would be nice to plot. I recognize that these are hard experiments, so if they are not available it is not absolutely necessary.

Reply: We thank the Reviewer for this helpful suggestion. We do have a few measurements extending to 2 ps that show no detectable signal, indicating that the signal decays to zero. However, as these measurements were only conducted a couple of times, the resulting error bars are quite large, which could lead to misinterpretation. For this reason, we opted to omit this data from the figure to maintain clarity. Thank you for your understanding, and we appreciate your attention to this detail.

4. Similarly, in Figure 4b, why not show spectra extending to longer times? Figure 4b goes to 500fs, but Figure 3 shows kinetics out to 1.25ps. Maybe the additional spectra could be put in the supplemental information, since too many spectra could take away from a very nice Figure 4.

Reply: We appreciate the Reviewer’s suggestion to extend the spectra in Figure 4b to longer times for comparison. For the kinetics shown in Figure 3, we fixed the XFEL beam energy and scanned the delay stage. In contrast, for Figure 4b, we fixed the time delay and scanned the XFEL energy using a monochromator. These two experiments were conducted separately and run until we achieved reliable statistics for each.

We do have one additional data point for Figure 4b at 1000 fs (see figure below), but due to low signal quality and poor statistics, we opted not to include it, as it holds limited scientific meaning in its current form. Based on our estimates, obtaining comparable error bars to those of the other time delays would require at least an additional 8–12 hours of beam time (approximately one shift), which was unfortunately beyond our available resources. Thank you for your understanding, and we hope this clarifies our decision.

Reviewer #4:

This paper reports very interesting FEL-XAS experiments on carrier dynamics in Au nanoparticles following plasmon excitation. Sample characterization by in-house methods is sound. The discussion is aided by ab-initio simulations. The FEL-XAS experiments per se were expertly conducted. The discussion contains interesting ideas and the paper, if published, will be of interest to researchers working in a very active area. However, I cannot recommend publication in its present form because the discussion and interpretation suffer from over interpretation in the part related to carrier dynamics (error bars too large, data points too small) and because there are very numerous ambiguities and cases of sloppy presentation. A detailed list is found below. It will be necessary to re-write the abstract, taking into account the comments below on the paper.

Reply: Thank you for your insightful and constructive feedback on our manuscript. We are pleased to hear that you found the FEL-XAS experiments, sample characterization, and ab-initio simulations well-executed and relevant to the field. We appreciate your recognition of the potential interest our findings may hold for researchers studying carrier dynamics in plasmon-excited nanoparticles. We acknowledge your concerns regarding over-interpretation in the discussion of carrier dynamics, particularly in relation to the robustness of the data given the size of error bars and number of data points. We agree that addressing this is essential for clarity and precision, and we are committed to revisiting our discussion to ensure it remains well-supported by the data. We will carefully review our interpretation of carrier dynamics and adjust our claims to reflect the uncertainties accurately. Furthermore, we appreciate your detailed list highlighting areas needing clarification and revision. We recognize the importance of clear, unambiguous presentation and will thoroughly revise the manuscript to address these issues. The abstract will also be rewritten to align with the clarified points in the main text. Thank you once again for your valuable feedback, which will undoubtedly improve the rigor and readability of our work. We look forward to submitting a revised manuscript that meets the journal's standards and your recommendations.

Major

• Line 100, Fig. 1D. The fits are not visible in Fig. 1D. You state that they are “single exponential”, but it seems the time dependence is more complex. See also later comment on lines 164 – 173.

Reply: We thank the Reviewer for highlighting this point. We agree that the decay dynamics involve more than a single component. Our intent was to convey that the short-lived component, associated with electron-phonon scattering, can be reasonably fitted with a single exponential decay. It is well established that in this time window, transient absorption spectroscopy (TAS) data are best fitted using a two-component model: a short-lived component for electron-phonon scattering and a longer-lived component for phonon-phonon scattering. We apologize for any confusion in our initial description. We have clarified this in the revised manuscript and have darkened the fitting lines for the two-component exponential model to improve visibility. The extracted electron temperature was determined using the single exponential fit, as specified.

Action taken: We have revised figure 1D to make fits more visible. We have updated the manuscript text with:

Revised figure caption:

(D) Excitation power-dependent bleach recovery dynamics. Kinetic traces extracted at 500 nm (horizontal dashed line in C) with double exponential fits, shown across varying laser fluences.

And have updated statement:

According to eq. 2, the ΔT_e was estimated for each laser fluence from the τ_{e-ph} extracted from the first exponential decay of the plasmonic resonance TAS data.

• Line 104. Fig. 1A does not seem correspond to the physical phenomena described in this section. It would be better to depict separately plasmon excitation, followed by electron – hole generation due to plasmon de-excitation and subsequent damping and thermal dissipation. Also, the description of the phenomena in lines 104 – 114 could be more streamlined and clearer. Check also the comment on line 252.

Reply: We thank the Reviewer for this helpful feedback. We have revised Figure 1A to improve clarity and ensure consistency with the physical phenomena discussed. The description of the process has also been streamlined to enhance readability. Given the broad range of timescales involved, it is indeed challenging to capture the entire process in a single, simplified figure. However, we believe the updated figure offers a clearer representation. Finally, we emphasize that the manuscript focuses specifically on the processes of hot carrier generation, multiplication, and relaxation, and we have made every effort to ensure these stages are accurately represented in both the figure and the text.

Action taken: We have revised Figure 1A and streamline the process description with the text below:

The mechanism for hot carrier formation and thermalization following localized surface plasmon resonance (LSPR) excitation is illustrated in Figure 1A. In brief, the light's electric field coherently excites the valence electrons in gold, and the subsequent decoherence of this plasmonic excitation leads to non-thermal electron distributions. This occurs via intraband transitions, often aided by phonon scattering or transitions from Landau damping and surface collisions, with a timescale of 10-100 fs. Initially, photon absorption produces an out-of-equilibrium electron energy distribution, which resembles a double-step function in electron occupancy. Over hundreds of femtoseconds, energy redistribution among the electrons progresses until a high-temperature Fermi-Dirac distribution is reached. Finally, electron-phonon interactions lower the electron temperature over the course of a few picoseconds. This process has been well-supported by theoretical models and confirmed experimentally through pump-probe optical spectroscopy, wherein a visible/near-infrared pump excites conduction band carriers, and a probe pulse tracks the time-dependent changes in transmission or reflectivity as the carriers generate and relax. The non-radiative decay of the plasmon resonance further relaxes through phonon-phonon scattering over hundreds of picoseconds, eventually releasing the generated heat to the surroundings over tens of nanoseconds. These latter stages are beyond the scope of this study.

• Line 164 – 167. You state that the increase in electron temperature was found by “the first exponential fit”. What does this mean? Also, the table in fig. 2B reports two relaxation times: e - ph and ph – ph. These are never described in the text. The equation used to fit the kinetic traces in Fig. 1D must be reported, indicating the fitting parameters.

Reply: We thank the Reviewer for this insightful observation. The TAS kinetics reveal two relaxation times for electron-phonon (e-ph) and phonon-phonon (ph-ph) interactions, as shown in Table 2B, which are now more fully described in the context of the overall decay dynamics. We also recognize the importance of presenting the fitting model clearly. Accordingly, we have added the equation used to fit the kinetic traces in Figure 1D to the supplementary information (Eq. S1). Thank you again for helping us enhance the clarity and completeness of our work.

Action taken: We have added the comment below to the manuscript text:

This term can be used to estimate the average electron temperature, as outlined in the following sections. The kinetic traces extracted at 500 nm were fitted with a double-exponential decay model to obtain the τ_{e-ph} for the short-lived electron-phonon scattering component and τ_{ph-ph} for the longer-lived phonon-phonon scattering component, as described in supplementary information.

And the fitting equation to the supplementary information:

The plasmonic electron-phonon lifetime (τ_{e-ph}) was extracted by fitting the decay of the bleach at 500 nm. The kinetic traces were fitted using a sum of convoluted exponentials, following the methodology detailed in previous publication, with the mathematical expression:

$$S(t) = e^{\left[-\left(\frac{t-t_0}{t_p}\right)^2\right]} * \sum A_i e^{\left(-\frac{t-t_0}{\tau_i}\right)} \quad (\text{eq. S1})$$

Where $t_p = \frac{IRF}{2\ln 2}$ and IRF is the width of the instrument response function (full width half maximum), t_0 is the time zero, A_i and τ_i are the amplitude and the decay times respectively, and * is the convolution operator.

• Lines 168 – 173. You state that the increase in electron temperature “stagnates”. Do you mean “saturates”? Any way, saturation is observed in Fig. 2C only for laser fluencies used in the TR-XAS experiments.

Reply: We thank the Reviewer for this comment. In the context we originally used the term, "stagnate" was intended to convey the same meaning; however, for clarity, we have replaced it with "saturate." While we agree that saturation is very noticeable in the TR-XAS data, we also observe a similar slowing in signal increase at higher fluences in the TAS data. This alignment allows for consistent fitting across TAS and TR-XAS, strengthening the connections between the two datasets.

• Line 123. You state that the LSPR is centered “nominally” at 520 nm. Why “nominally”? In Fig. S3 the peak of the resonance appears to be at 550 – 560 nm. Is this the effect of the sloping background? Can you be more accurate in the determination of the energy of the LSPR? This would help in justifying the choice of 532 nm for the pump probe in the XAS measurements (line 230).

Reply: We thank the Reviewer for this comment. We have removed the term in question, as it did not add clarity to the text. Additionally, we have replotted the data to reflect the measurements directly, clearly showing the LSPR maximum at 520 nm. We acknowledge that the figure in the unrevised SI may have been misleading; this was partly due to a prior request to remove the sloping background. We believe that plotting the data as measured provides the most accurate representation and have adjusted the figure accordingly. Thank you for helping us enhance the accuracy of our presentation.

Action taken: We have replotted the optical absorption data for the colloidal Au nanoparticles used in the study (Fig. S3), without any additional processing, which clearly shows the LSPR maximum at 520 nm. Please note that the UV-Vis spectrum was obtained from the colloidal solution immediately after synthesis, whereas for the TR-XAS measurements, the sample was diluted fivefold.

• Line 225 and Fig. S6. The NP size for all intermediate cases should be reported. It is surprising that the DOS curves do not change gradually. The one in the middle is highly structured. Check.

Reply: We thank the Reviewer for this insightful comment. In the original figure, we included sizes calculated with various shapes, which, as you noted, were not properly labeled. Since the preferred geometry for our study is icosahedral, we have revised the figure to include only icosahedral geometries and have clearly labeled the sizes in each panel. This updated figure more accurately demonstrates our intended observation in Figure S6: that with icosahedral geometry, the DOS begins to resemble the bulk state even at small sizes (above 2 nm). The previous mid-panels, which displayed highly structured DOS associated with shapes that have a high exposed surface and defect-dominated states, were not relevant to our study's focus

and have therefore been removed. Thank you again for helping us improve the clarity of our presentation.

Action taken: We have replotted the calculate DOS of Au icosahedral shape (Fig. S6) only.

• Line 248. The changes observed are presumably due to changes in the occupation of states not to modification of the density of states. These are different concepts.

Reply: We thank the Reviewer for this helpful clarification. We agree with the comment and have updated the text accordingly.

• Line 252. The formation of holes is presumably due to the de-excitation of the plasmon following its optical excitation. The present wording is confusing. This is linked to the ambiguous phrasing in lines 104 -114 and Fig. 1A.

Reply: We thank the Reviewer for this helpful clarification. We agree with the comment and have updated the text accordingly.

Action taken: We have updated the text with:

Thus, the positive signal observed below the Au E_F is attributed to the formation of a hot hole population from the non-radiative decay of the plasmon. Conversely, the negative signal observed above the Au E_F is due to hot electrons filling empty states, as expected.

• Lines 268 – 274, Fig. S8 and its caption. This is very unclear. What is the relation between the plot of ΔA vs. fluence values and the electron temperature? How was the conclusion that the electron temperature is constant at about 2000 K reached (Fig. 2 C and caption to Fig. 8)? What is the meaning of the straight line fit in fig. 8? I have serious reservations on fitting the highly scattered points (no error bars) with a straight line, any function can pass through those points. Finally, it is stated that at 98 mJ/cm² there is no saturation, but the data in Fig. 2C seem to contradict this.

Reply: Thank you for your thorough feedback. We understand that the points you raised require clarification, and we appreciate the opportunity to address them in detail.

- 1. Relation between ΔA vs. fluence and electron temperature:** The plot in Fig. 2C presents electron temperature on the y-axis, estimated using equation 2. Given that the TAS ΔA at the early stages reflects only the hot carrier dynamics, and the ΔA in the TR-XAS also exclusively represents hot carriers, we can unify both measurements using electron temperature as the common y-axis and fluence as the x-axis. This approach allows us to establish a clear correlation between fluence and electron temperature through controlled fluence variation and connect TAS with TR-XAS measurements. In practice, the plot can be viewed as a straightforward ΔA versus laser fluence, with ΔA converted into electron temperature to provide a more physically meaningful interpretation of the process.
- 2. Electron temperature saturation at ~2000 K:** The conclusion that the electron temperature saturates at ~2000 K comes from the best fitting of the TAS and TR-XAS ΔA s, which levels off at ΔA equivalent to 2000 K.
- 3. Straight line fit in Fig. S9:** We agree that the original figure may have been somewhat misleading. Our intention was to demonstrate that the slope of the TR-XAS ΔA versus laser fluence is very shallow, especially when compared to the TAS ΔA versus laser fluence. This shallow slope in the TR-XAS signal suggests that it remains largely constant, indicating electron temperature saturation. Consequently, any fluence within this range is suitable for TR-XAS measurements, and we selected a mid-range value of 98 mJ/cm² for our experiments. The revised figure, now with all relevant data included, clearly shows this signal stagnation above 5 mJ/cm². Specifically, the TR-XAS slope is

very small (approximately 12), especially when compared to the TAS slope (approximately 380, as shown in Figure S10).

Thank you again for your insights. These clarifications should improve the clarity and consistency of our discussion.

Action taken: We have added a trendline to Figure S9 (formerly Figure S8) and introduced Figure S10 for comparison.

We have revised the text with:

An important consideration in the TR-XAS experiments is the pump laser fluence. Figure S9 presents the fluence dependence of the TR-XAS signal at an incident energy of 11916 eV with a pump-probe delay of 100 fs. While TR-XAS ΔA at 100 fs shows an increase with rising laser fluence, the rate of this increase is relatively modest (slope = 11.1 ± 3.5), particularly in comparison to the TAS ΔA versus laser fluence slope 383 ± 95 (see Figure S10). This indicates that the observed rise in TR-XAS ΔA does not significantly alter the overall electron temperature. Consequently, the TR-XAS signals remain within the ~ 2000 K electron temperature plateau, achieved at approximately 5 mJ/cm^2 of laser excitation. Thus, to balance optimal signal intensity with minimal saturation effects, a fluence of 98 mJ/cm^2 was selected for the TR-XAS measurements.

• Lines 275 – 277. There is an apparent contradiction. First, “generation of hot carriers through LSPR decay”. Then “the choice of pump energy excludes direct vertical transitions as a significant decay channel”. Rephrase, otherwise it is incomprehensible.

Reply: We thank the Reviewer for this helpful clarification. We agree with the comment and have updated the text accordingly.

Action taken: We have updated the text with:

The TR-XAS signal directly indicates hot carrier generation via LSPR non-radiative decay (Fig. 2D). The chosen laser pump energy minimizes interband excitation, ruling out direct vertical interband transitions as a significant decay pathway.

• Line 336. You state that the “thermalization time aligns with the measured value when exciting solely the LSPR transition”. What does this mean? Is there another excitation besides LSPR?

Reply: We thank the Reviewer for this helpful clarification. We agree with the comment and have updated the text accordingly.

Action taken: We have updated the text with:

The estimated thermalization time for hot carriers is consistent with previous studies where LSPR transitions are excited without significant interband excitation and the predictions of the classic two-temperature model.

• Line 380. Again, there seems to be confusion in the use of “excitation” and “de-excitation”. It is the de-excitation of plasmons which leads to the variation of states above and below the Fermi level.

Reply: We thank the Reviewer for this helpful clarification. We agree with the comment and have updated the text accordingly.

Action taken: We have updated the text with:

Figure 4B illustrates the temporal evolution of the hot carrier population and their energy distribution, resulting from the non-radiative decay of optically excited LSPR transitions. As

expected, this non-radiative deexcitation of the plasmon depopulates states below the Fermi energy and populates states above it.

• Line 384, Fig. 4C. There are 3 experimental points for “population” and “distribution”. How were these determined, what is the meaning? Anyway, the error bars are very big, I do not believe it can be stated that they reach a maximum at different times, there is too much scattering of the data points; I think this is an over interpretation of the data.

Reply: We thank the Reviewer for the thoughtful comment. The hot hole population was estimated by integrating the area under the peak at each delay time, while the energy distribution is assessed based on the peak’s width. We acknowledge that individual data points show some noise; however, these two parameters are derived using all data points collectively, resulting in improved statistical reliability. Moreover, even a straightforward inspection of the curves in Fig. 4B demonstrates that the carrier population and their energy distribution do not peak at the same time. Thus, we respectfully disagree with the concern that the data may be over-interpreted.

• Line 388, Fig. 4A. It is misleading to indicate the photon energy as a single vertical line. The photon induces a transition, I suggest a horizontal arrow.

Reply: We thank the Reviewer for this insightful comment and completely agree that the optical pulse has a width (± 0.13 eV in this case), allowing for the excitation of various transitions within the pulse envelope. This effect is represented in our figure; however, due to the x-axis scale, it may appear as a single line. To clarify, we have updated the figure caption to emphasize this detail.

Action taken: Figure 4A caption was updated with:

The optical pulse in green is depicted as a Gaussian band centered at 2.33 eV with a full width at half maximum (FWHM) of 15 nm, corresponding to an energy envelope of ± 0.13 eV

• Line 394, Fig. 4D. What is the meaning of the curves? The ones for -2.8 eV have a surprising undulation! Why is the energy of the holes “affected” by the Au core – hole lifetime? Do you refer to broadening? This is different.

Reply: We thank the Reviewer for the helpful comment. This figure is intended to demonstrate that carriers with different energies exhibit distinct decay behaviors, with those at intermediate energies showing oscillations due to the coexistence of Auger heating and the predominant impact excitation. If only impact excitation was present, we would expect a smooth decay, with higher-energy carriers decaying more rapidly. Unfortunately, the current statistics and energy resolution do not allow us to capture this aspect in greater detail. However, we plan to use this observation as a basis for requesting XFEL beamtime to measure the signal with higher energy resolution using a von Hamos dispersive spectrometer.

• Lines 398 – 406. This paragraph and Fig. S9 are unnecessary. The particles used are 7 nm with small dispersion, there are no size effects. The same message has already been given in Fig. S6, why the repetition? In the caption: projected on what? Why is the photon energy reported as a vertical line? A photon induces a transition, it should be represented as a horizontal line, as already mentioned.

Reply: We thank the Reviewer for the comment. This paragraph was added in response to a previous revision request. We have now condensed it and removed Fig. S9. However, we are willing to remove the paragraph entirely if the editor agrees.

• Line 407 onwards. As already mentioned in line 384, I find all this discussion on the width and population of electrons and holes highly speculative and an over interpretation of the data. Error bars are too big and the number of points too small to be credible. At most, a qualitative interpretation might be acceptable.

Reply: We thank the Reviewer for the helpful comment. As mentioned previously, our analysis includes all data points, making the population analysis statistically significant. However, we wish to emphasize that our energy distribution analysis is intended only as a qualitative assessment. We clearly acknowledge the need for higher energy resolution to make quantitative statements about carrier energy.

• Line 407: higher sensitivity to the formation of hot holes with respect to what? Hot electrons? Why? In line 408 why “however”? Is this an explanation of the asymmetry?

Reply: We thank the Reviewer for this helpful clarification. We have updated the text to make this clearer.

Action taken: We have updated the text with:

The signal asymmetry between hot electrons and holes can be partly attributed to the higher sensitivity of the L_3 -edge XANES transition to the formation of empty states in the d-shell, i.e., the hot holes. However, the shape asymmetry between hot electrons and holes is also anticipated because of the difference in electron and hole density of states, consistent with experimental and theoretical reports.

• Line 415 and Fig. S10. The DFTB+ simulations are interesting and apparently provide support to the interpretation. However, details on the simulations should be provided (in the SI). Simulations for various NP sizes are unnecessary (see also comment above on line 398). Intermediate times should be provided in the figure, not just the minimum and maximum. The ordinate scale for the biggest NP should be changed in order to make the figure intelligible. Rather than the number of atoms in the NP their size should be given, in order to make the presentation consistent. In line 425 the simulations are referred to as reporting “localization”. Why? What do you mean?

Reply: We thank the Reviewer for the comment. We have highlighted in the Supplementary Information the approach used for the simulations and referenced a paper detailing the experimental methods. While the simulations are preliminary, they suggest a greater localization of electrons compared to holes, which favors carrier multiplication. This process increases the carrier count while reducing their individual energy, helping to explain the observed asymmetry. At present, we cannot expand further, but this insight aids in rationalizing the observed asymmetry.

• Line 454. The discussion of hole and electron dynamics (Figs. 4C and S12) must be improved. In Fig. S12 the intensity for three energy values actually increases at 500 fs; however it is just one data point. Is it credible? What is the expected behavior for a Fermi Dirac distribution?

Reply: thank the Reviewer for this comment. We agree that the statistical strength of the hot electron signal is limited and does not support definitive conclusions. However, we believe that the observed changes for the hot holes are significantly larger than the margin of error, allowing us to make the qualitative statements included in this section. This analysis was initially requested by a previous Reviewer, and while it provides additional context to the data in Figure 4C, it is not intended as a standalone result with substantial relevance.

Minor/formatting/language

- Lines 34 – 35. Surface plasmons have been known for decades and are a textbook subject. They have not “emerged” (it is implied, recently). Rephrase.
- Line 36. Why “distant” radiation? Distant from what?
- Line 37. Instead of “profound” use “high” or “significant”.
- Line 38. Instead of “attributes” use “characteristics”.
- Line 59. Instead of “interplay” you probably mean “interaction”. Anyway, use of XAS to investigate radiation – matter interactions is besides the point in this context, why quote it? You are using XAS to probe the evolution of the electronic structure following plasmon excitation, focus on that.
- Line 62. Following x-ray absorption core electrons undergo an excitation and perform a transition, they do not “shift”.
- Lines 65 – 66. This phrase is completely out of context.
- Line 69. Carrier “participation” in what? In what process?
- Line 81. I suggest to quote some other FEL – XAS studies of functional materials, in order to better represent recent activity in this field. For example: Y. Uemura et al., *Angew. Chem.* 55, 1364 (2016). Y. Obara et al., *Struct. Dyn.* 4, 044033 (2017); Y. Uemura et al., *Chemical Communications* 53, 7314 (2017); Pelli Cresi et al., *Nano Lett.* 21, 1729–1734 (2021).
- Line 125. The UV-Vis spectra (add).
- Line 134. What are “typical” lase fluences? Typical of what?
- Line 143. Instead of “variance” I think you mean “difference”. The variance refers to a distribution or population, not to two quantities.
- Line 154. Check the signs in Eq. 1. I suspect there should be a + sign on the RHS of the second equation.
- Line 179. What do you mean in stating that XAS allows tracking hot carrier “energetics”? Maybe hot carrier energy distribution?
- Line 184. I presume that the optical pulse is not “delayed” with respect to the X-ray one, rather the opposite. The time interval between the optical pulse and the X-ray one is varied.
- Line 193. It is Fig. S4.
- Lines 196 – 199. The absorption edge (i.e. the discontinuity) of the Au L3 edge is clearly visible. It is the white line which is greatly damped (with respect to Pt) because of the d10 electronic structure of Au. It is well known that XAS at metal L3 edges is sensitive to empty states in the 5d shell, so use “illustrating” rather than “revealing”, it is hardly a new discovery.
- Lines 218 – 219. What do you mean by “genuinely describe the Au NPs used in electronic structure”? Used in electronic structure? A single FEL – XAS spectrum does not “uphold” the ability to capture transient electronic structure. Rephrase, write more clearly.
- Line 237. Instead of “largely excludes” use “excludes to a high degree” or similar.
- Line 243. Corroborating the presence of light induced
- Line 248. The changes observed are presumably due to changes in the occupation of states not to modification of the density of states. These are different concepts.
- Line 252. Instead of “forming” use “formation”.
- Line 307. Define the symbol $\Gamma(\text{hom})$. Homogeneous?

Reply: We sincerely thank the Reviewer for the careful reading of our manuscript and for providing detailed corrections minor issues.

Action taken: We have revised the manuscript text to incorporate the suggested statements and have added the recommended references pertaining to XAS-FEL studies.

Answers to the Reviewers comments:

Reviewer #1:

The authors have addressed my concerns. The paper can be published.

Reply: Thank you for your time and thoughtful review of our manuscript. We sincerely appreciate your positive feedback and support for its publication.

Reviewer #2:

My main criticism of this paper was that it is fundamentally flawed as the authors have not performed control experiments.

Reply: We sincerely appreciate the Reviewer's time and effort in evaluating our manuscript. However, we are concerned that the Reviewer has not acknowledged the substantial additional control experiments incorporated in response to earlier feedback.

In the initial revision, we added extensive power-dependence measurements at 532 nm, followed by both optical and X-ray probes. These results are included in the manuscript (Figure 2C) and presented in full in Figures S9 and S10.

Additionally, we provided power-dependence data concerning the X-ray probe (Figure S8). By comparing transient XAS measurements at a 100-fs time delay under two different X-ray flux conditions, we demonstrated that the signal remains unaffected by X-ray pulse intensity. While we acknowledge that only two power levels were tested, the fluence range between them is significantly larger than any expected variations due to pulse intensity jitter. Further, any nonlinear X-ray interaction would square the signal intensities.

Further, in our most recent revision, we complemented our dataset with optical transient measurements following excitation at different wavelengths, showing that excitation conditions influence the observed dynamics. This aspect is further discussed below.

Given these extensive additional experiments, we are uncertain about what further control studies the Reviewer envisions to address the non-radiative plasmonic decay in gold nanoparticles. We would greatly appreciate specific guidance on any remaining concerns.

As an example, I pointed out that the pump-probe results at 520 nm can be easily due to the interband excitation in which case the whole argument of the authors about plasmon-assisted effect falls apart.

Reply: We appreciate the Reviewer's comment; however, the statement contains two inaccuracies.

First, all XFEL pump-probe data were collected using **532 nm excitation**, not 520 nm. The 520 nm wavelength referenced by the Reviewer corresponds to the localized surface plasmon resonance (LSPR) maximum, not the excitation wavelength used in our experiments.

Second, our choice of **532 nm excitation**, is intentionally **red-shifted from the LSPR maximum**, to **minimize interband contributions**. This approach is supported by prior studies, including Lyu et al. (*J. Phys. Chem. C* **127**, 15685–15698 (2023)), which illustrate that excitation just to the red of the LSPR peak significantly reduces interband effects (see schematic below). Furthermore, the Au d-band onset is located at 2.5–2.58 eV (~496–480 nm), as reported by Johnson et al. (*Phys. Rev. B* **6**, 4370–4379 (1972)) and Dreesen et al. (*Appl. Phys. B* **74**, 621–625 (2002)). This positioning means that at 532 nm (2.33 eV), interband transitions contribute minimally, reinforcing our plasmonic interpretation.

Action taken: text clarification.

Since completely avoiding interband excitation when exciting close to the LSPR maximum is not feasible, it was crucial to demonstrate that its contribution to the overall signal remains minimal when exciting to the red of the LSPR peak. To investigate this, optical TAS measurements were performed at different excitation wavelengths: 450 nm (predominantly interband excitation, below the LSPR peak), 520 nm (resonance excitation, at the LSPR peak maximum), and 532 nm (intraband excitation, above the LSPR peak). A comparison of the kinetic traces extracted near the excitation wavelength (Fig. S7) reveals that pure interband transitions (450 nm) are significantly less efficient than intraband transitions (532 nm) in generating hot carriers. Additionally, the observed decrease in signal amplitude when exciting beyond the LSPR peak, compared to excitation at the LSPR maximum, further supports the conclusion that interband excitation has a diminished contribution when excitation wavelengths are chosen to the red of the LSPR maximum.

To address this issue the authors have presented transient measurements results for different pump wavelengths and claim that these results prove that the excitation at LSPR wavelength are not dominated by interband excitation. I strongly disagree with this assertion. The difference in the amplitude of the transient signals in Fig S7 are most likely due to plasmon near-field enhancement of interband absorption. The lifetime variation in Fig. S11 is most likely due to thermal effects (electron-phonon coupling depends on heat capacity which is a function of the temperature). The authors can easily verify this by performing power-dependent measurements at LSPR resonance and observe longer decay times at higher excitation power densities. Furthermore, they cite Phys. Rev. B 50, 15337-15348 (1994) to support their claim, however the pump wavelength in that work was in the infrared (~900 nm) and the observed decay was simply due to electron-phonon thermalization and not plasmon decaying.

Reply: We appreciate the Reviewer's comment and careful analysis; however, we believe the discussion results from the non-intentional Reviewer's overinterpretation. Our primary goal with the TAS experiments at variable excitation wavelengths was to demonstrate that both the amplitude and kinetics of the signals change significantly when the interband contribution varies, supporting our assertion that the observed dynamics at 532 nm excitation are not dominated by interband transitions.

The Reviewer notes that the plasmon excitation could enhance the interband contribution due to near-field effects, which implies that LSPR excitation occurs at 532 nm for such enhancements to be possible. However, this does not imply that enhancement of the interband excitation results in its dominance of the response at 532 nm. The key question is to estimate the relative contributions of plasmon and interband-driven processes. We emphasize that the Au d-shell onset is located at 2.5–2.58 eV (~496–480 nm), as reported by Johnson et al. (Phys. Rev. B 6, 4370-4379 (1972)) and Dreesen et al. (Appl. Phys. B 74, 621-625 (2002)). Since our excitation wavelength is 532 nm (2.33 eV), the number of available states for direct interband transitions is significantly reduced. Even with potential near-field enhancement, the probability of interband excitation remains low at the used photon energy, resulting in the contribution from interband absorption being at the level of the background signal.

Regarding the Reviewer's suggestion about power-dependent measurements at LSPR, we agree that such experiments are key, which is why we performed the power-dependence studies that are included in the manuscript (Figure 2C) and presented in full in Figures S9 and S10. It is clear, as the Reviewer, states that increasing the excitation power extends the electron-phonon lifetime and consequently raises the electron temperature, in accordance with equation 2 until reaching a plateau behavior. At this point, the electron temperature reaches approximately 2000 K, consistent with previous predictions.

As mentioned by the Reviewer, Sun et al. (Phys. Rev. B 50, 15337–15348 (1994)) utilized an excitation wavelength of 900 nm. However, this approach significantly reduced the signal intensity, necessitating long acquisition times. While such an approach is feasible with laboratory instrumentation, it is impractical for XFEL experiments due to limited beam availability and the highly competitive nature of facility access. Consequently, we selected an

excitation wavelength that predominantly excites the plasmonic response while acknowledging that some interband contributions may occur. However, their overall impact on the observed signal remains minimal. The simple fact that our dynamics are similar to Sun's suggests that plasmonic excitation is the main culprit for the signal.

Once again, we appreciate the Reviewer's suggestion. However, the existing evidence strongly suggests that interband transitions do not dominate the signal at 532 nm, even when considering plasmonic near-field effects.

Action taken: text added to the discussion to convey that mainly plasmons are excited.

As a final remark, it is important to reiterate that while the excitation wavelength used for transient XANES may induce a small fraction of interband transitions due to plasmonic near-field enhancement, our experimental design was carefully chosen to minimize this effect. By selecting an excitation wavelength slightly red-shifted from the LSPR maximum and positioned relative to the expected Au d-band onset, we effectively reduce the likelihood of significant interband contributions. The validity of this approach is further supported by the similar dynamics reported by Sun et al., who utilized 900 nm excitation to completely suppress interband transitions. Therefore, the collective evidence strongly indicates that interband transitions do not play a dominant role in the observed signal at 532 nm, even when accounting for potential near-field enhancements.

Reviewer #3:

The authors have done a nice job addressing my comments/concerns in the manuscript and in their responses. I have no further requests/questions. These are very difficult experiments on a beamline that has limited access. Overall, I believe the readership of Nature Communications will find this work very interesting.

Reply: We sincerely appreciate your thoughtful review and positive feedback on our manuscript. Your recognition of the challenges associated with these experiments and the value of our work is highly encouraging. Thank you for your time and support for the manuscript publication.

Reviewer #4:

The authors have responded in detail and in a satisfactory manner on my comments to the first submitted version. They have revised the paper accordingly. Fine by me.

Reply: Thank you for your thorough review and positive assessment of our revisions. We are grateful for your endorsement for publication.

Answers to the Reviewers comments:

Reviewer #1:

I feel the paper can now be published. It is true that the debate about the contribution of d-transitions vs plasmon excitation is sound and raises a number of questions. However, overall it is fair to assume that even if d-excitation is present, the bulk of the observations comes from plasmon excitation. In addition, the results in this paper will surely stir further studies that will look more carefully into this issue.

Reply: Thank you for taking the time to carefully review our manuscript and for your endorsement. We truly appreciate your thoughtful feedback and your recognition of the broader impact of our findings.

We agree that the debate regarding the contributions of d-transitions versus plasmon excitation is an important one, raising many intriguing questions. While d-excitation may play a role, we concur with your assessment that the primary observations in our study are driven by plasmon excitation. We also share your perspective that these results will inspire further investigations into this topic, helping to refine our understanding of the underlying mechanisms.

Once again, we sincerely appreciate your valuable insights and support.

Reviewer #2:

I feel that, at this point, we are unfortunately starting to go in circles. As I mentioned in my initial review, the relevant control experiment would involve conducting the same measurements (visible pump, X-ray probe) on bulk gold samples. This approach would isolate the contribution of plasmon dephasing. The power-dependent measurements presented by the authors do not constitute proper control experiments for a simple reason: they do not address the primary issue of this experiment—the potential for interband absorption at 532 nm.

The authors' claims rest on the assumption that interband absorption in gold can be neglected at 532 nm. Based on my experience and the extensive literature on this subject that I am familiar with, this assumption is not accurate.

Unfortunately, I am unable to provide further constructive feedback on this paper. I understand that these measurements are very challenging, and there may be little the authors can do at this stage. I sympathize with their efforts; however, I cannot overlook the fact that the experimental design appears fundamentally flawed, in my opinion.

Reply: Thank you for taking the time to carefully review our manuscript and for your detailed feedback. We acknowledge that at this stage, the discussion is reaching an impasse, and it may not be possible to arrive at a fully unified conclusion on this matter.

We recognize that some interband transitions can be excited at 532 nm, and our statement acknowledges this possibility. However, based on the extensive literature on the position of the Au d-band relative to the Fermi level, along with our transient absorption spectroscopy (TAS) measurements, we believe that interband absorption is unlikely to play a significant role in our observations.

Regarding your suggestion for a control experiment involving the same measurements (visible pump, X-ray probe) on bulk gold samples, we revisited our records but did not find this recommendation in the previous reviews. If it had been explicitly suggested earlier, we would have been able to clarify this point sooner. Nevertheless, we can confirm that Au bulk excitation with the pump fluence used in our experiment showed no detectable signal. However, if the fluence is significantly increased and the pulse duration is shortened, multi-photon excitation can occur, leading to electron ejection and modification of the Au density of states (DOS). We observed this effect in related experiments using attosecond probe pulses to examine Au valence states.

We appreciate your acknowledgment of the challenges involved in these measurements and your thoughtful engagement with our work. While we may not fully agree on all aspects, we value the constructive discourse that this review process has fostered.

Reviewer #4:

The authors have done a good job of responding in detail to all comments, especially those of referee 2. In my view, the paper should be published and will contribute significantly to research in this topical field.

Reply: Thank you for taking the time to carefully review our manuscript and for your endorsement. We truly appreciate your thoughtful feedback and your recognition of the broader impact of our findings.